



# Volcanic SO₂ Effective Layer Height Retrieval for OMI Using a
# Machine Learning Approach
Nikita M. Fedkin[1], Can Li[2], Nickolay A. Krotkov[2], Pascal Hedelt[3], Diego G. Loyola[3], Russell R.
Dickerson[1], Robert Spurr[4]
1: Department of Atmospheric and Oceanic Science, University of Maryland, College Park, MD, USA
2: NASA Goddard Space Flight Center, Greenbelt, MD 20771, USA
3: German Aerospace Center (DLR), Remote Sensing Technology Institute (IMF), Oberpfaffenhofen, Germany
4: RT Solutions Inc., Cambridge, MA, USA
*Correspondence to*: Nikita M. Fedkin (nfedkin@umd.edu)
**Abstract.** Information about the height and loading of sulfur dioxide ($SO_2$) plumes from
volcanic eruptions is crucial for aviation safety and for assessing the effect of sulfate aerosols on
climate. While $SO_2$ layer height has been successfully retrieved from backscattered Earthshine
ultraviolet (UV) radiances measured by the Ozone Monitoring Instrument (OMI), previously
demonstrated techniques are computationally intensive and not suitable for near real-time
applications. In this study, we introduce a new OMI algorithm for fast retrievals of effective
volcanic $SO_2$ layer height. We apply the Full Physics Inverse Learning Machine (FP_ILM )
algorithm to OMI radiances in the spectral range of 310-330 nm. This approach consists of a
training phase that utilizes extensive radiative transfer calculations to generate a large dataset of
synthetic radiance spectra for geophysical parameters representing the OMI measurement
conditions. The principal components of the spectra from this dataset in addition to a few
geophysical parameters are used to train a neural network to solve the inverse problem and
predict the $SO_2$ layer height. This is followed by applying the trained inverse model to real OMI
measurements to retrieve the effective $SO_2$ plume heights. The algorithm has been tested on
several major eruptions during the OMI data record. The results for the 2008 Kasatochi, 2014
Kelud, 2015 Calbuco, and 2019 Raikoke eruption cases are presented here and compared with
volcanic plume heights estimated with other satellite sensors. For the most part, OMI-retrieved
effective $SO_2$ heights agree well with the lidar measurements of aerosol layer height from Cloud-
Aerosol Lidar and Infrared Pathfinder Satellite Observations (CALIPSO) and thermal infrared
retrievals of $SO_2$ heights from the infrared atmospheric sounding interferometer (IASI). The
errors in OMI retrieved $SO_2$ heights are estimated to be 1-1.5 km for plumes with relatively large
$SO_2$ signals (> 40 DU). The algorithm is very fast and retrieves plume height in less than 10 min
for an entire OMI orbit. This approach offers a promising prospect of using physics-based
machine learning applications to other instruments.





**1 Introduction**


The observation and tracking of emissions from volcanic eruptions are crucial for both air traffic
safety and for assessing climate forcing impacts from volcanic sulfate aerosols. In the last 10
years, volcanoes have emitted roughly 20-25 million metric tons of sulfur dioxide ($SO_2$) per year
through passive degassing (Carn et al, 2017). Explosive volcanic eruptions, however, can
additionally release large $SO_2$ amounts high into the atmosphere. $SO_2$ can be converted to sulfate
aerosols within 2-3 days in the troposphere (Lee et al., 2011) and within a few weeks in the
lower stratosphere (von Glasow et al., 2009, Krotkov et al., 2010). Sulfate aerosols are known to
have a cooling effect on climate, especially if an $SO_2$ plume is injected into the lower
stratosphere and remains there for longer periods of time. This is demonstrated by significant
eruptions such as Mt. Pinatubo in 1991 that temporarily reduced global temperatures by up to
0.5℃ (McCormick et al, 1995). Aside from releasing $SO_2$, volcanoes also emit large amounts of
ash into the atmosphere which can have adverse impacts on air travel. Ash from volcanic plumes
can often interfere with flight paths, greatly reduce visibility near the ground, and cause damage
to the aircraft including engine failure (Carn et al., 2009). In addition, $SO_2$ causes sulfidation in
the engines, an effect that can reduce their lifetimes in the long term. From 1953 to 2009, over
120 aviation incidents involving volcanic activity were reported, with roughly 80 of them
involving serious damage to the airframe or engine (Guffanti et al., 2010). There is also the
possibility of highly concentrated volcanic $SO_2$ plumes producing acidic aerosols which can
cause irritation of the eyes, nose and respiratory airways of occupants inside airplanes (Schmidt
et al., 2014). In many cases $SO_2$ and ash are often collocated, thus making estimates of $SO_2$ layer
height very useful for aviation hazard mitigation and volcanic plume forecasting. Lastly, the
accurate determination of $SO_2$ height can ideally aid in producing accurate $SO_2$ VCD estimates
given that those retrievals typically use a fixed *a priori* vertical distribution of $SO_2$ in the absence
of additional information on $SO_2$ height.
With remote sensing, these volcanic plumes can be regularly observed from space. In
particular, hyperspectral spectrometers such as the Ozone Monitoring Instrument (OMI),
GOME-2, OMPS, TROPOMI and others, have provided frequent and increasingly accurate
observations of global $SO_2$ amounts, through retrieval algorithms from backscattered radiance



measurements. The OMI instrument, a Dutch-Finish contribution to the NASA Aura satellite,
has been operational since 2004. OMI has 60 cross track positions (rows) and has a $13 \times 24$ km$^2$
spatial resolution at the nadir position (Levelt et al., 2006). The instrument uses two UV channels
and one visible channel to measure backscattered radiances from the Earth's atmosphere. In
general, $SO_2$ slant column amounts are retrieved from these measurements through the
differential optical absorption spectroscopy (DOAS) technique and then converted to vertical
columns using Air Mass Factors (AMFs). The 310.5-340 nm range in OMI's UV2 channel is
used in retrieving $SO_2$, with focus on the 310.8 and 313 nm wavelengths. The band residual
algorithm (Krotkov et al., 2006) and the Linear Fit (LF) algorithm (Yang et al., 2007) were first
used as the OMI operational algorithms for retrieving planetary boundary layer (PBL) $SO_2$ and
volcanic $SO_2$ vertical column densities (VCDs) respectively. These were replaced with the
principal component analysis (PCA) based algorithm (Li et al., 2013) which retrieves $SO_2$
amounts directly from spectral radiance measurements. The same technique was also applied to
OMI volcanic $SO_2$ retrievals (Li et al., 2017). This data-driven approach does not rely on
extensive radiative transfer modeling and has led to reduced biases and significant improvements
(Fioletov et al., 2015). For volcanic retrievals, algorithms still have uncertainties in $SO_2$ mass in
volcanic plumes, especially in the presence of relatively larger errors in the assumed *a priori*
profiles.

86        In addition to column amounts, backscattered radiances can also provide important

information about the height of an $SO_2$ layer. Conceptually, a change in altitude of an $SO_2$ plume
alters the number of backscattered photons going through the layer. If a plume is high in the
atmosphere, more photons that are scattered below the layer pass through the absorbing $SO_2$
plume. This results in larger $SO_2$ absorption structures in the measured radiance spectra,
especially in the 310-320 nm range where Rayleigh scattering is dominant. Relative to the $SO_2$
amount, obtaining a fast retrieval of the height of a volcanic plume presents a greater challenge.
Until recently, retrieval techniques have involved a direct spectral fitting approach that use BUV
measurements in conjunction with extensive forward radiative transfer modeling. For instance,
the Iterative Spectral Fitting (ISF) algorithm (Yang et al., 2009) for OMI was utilized to
determine the altitude of $SO_2$  layer by adjusting the height while minimizing the differences
between measured radiances and forward RT calculations. Another study has utilized an optimal
estimation algorithm along with the VLIDORT radiative transfer (RT) model to retrieve $SO_2$


density and plume height from the GOME-2 instrument (Nowlan et al., 2011). Sulfur dioxide
amounts and plume heights have also been estimated with the infrared atmospheric sounding
interferometer (IASI), through brightness temperature changes and relative intensities of
absorption lines (Clarisse et al., 2008; Clarisse et al., 2014). For these techniques, extensive
radiative transfer modeling is needed, in addition to a variety of assumptions including a
reasonable first guess for the plume altitude. Newer schemes were later developed for GOME-2
using the SOPHRI algorithm (Rix et al., 2012), a DOAS based technique that included
minimizing differences between plume height from simulated spectra and the assumed height
from measured spectra. This technique allowed for reasonably fast retrievals that could be used
in near real-time, thanks to the use of pre-calculated GOME spectra that are stored in a look up
table classified according to $SO_2$ column, $SO_2$ heights and other physical parameters. An even
faster and more efficient method for GOME-2 (Efremenko et al., 2017) and TROPOMI (Hedelt
et al., 2019) has made use of machine learning algorithms, specifically neural networks (NNs), to
develop a trained full physics inverse learning machine (FP_ILM) for retrieving $SO_2$ plume
height. This approach has shown good accuracy and speed fast enough for near-real-time
operations. The FP_ILM has also been used for retrieving ozone profile shapes (Xu et al., 2017)
and geometry-dependent Lambertian equivalent reflectivity (Loyola et al., 2020). The primary
advantage of this approach is the execution speed. By separating the training phase, which
involves large amounts of time consuming radiative transfer computations and machine learning
model training, from the application phase, the desired parameter can be retrieved within
milliseconds for a single satellite ground pixel using the inverse model. However, similar
methods of retrieving $SO_2$ layer height have not yet been implemented for OMI. Now in this
study, the FP_ILM has been applied to OMI to estimate $SO_2$ layer height from backscattered
earthshine radiance measurements. The retrieval was tested on four past volcanic eruption cases
and performance was assessed through machine learning metrics, as well as comparisons to other
datasets such as those from TROPOMI, IASI and CALIOP lidar instruments.
**2 Methodology:**
In general, the FP_ILM approach consists of two parts, the training phase and the application (or
operational) phase. The training phase starts with the generation of a synthetic training dataset of





top of the atmosphere (TOA) reflectance spectra from a radiative transfer model. This spectral
dataset is then used to train a Multi-Layer Perceptron Regression (MLPR) NN model to predict
the $SO_2$ layer height as an output. In the application phase, the trained inverse model is applied to
real OMI radiance measurements. This inverse model is optimized from the training, and the
predictions of $SO_2$ layer height based on the model are very fast as compared with the time-
consuming RT calculations during the training phase. The main steps of the algorithm are shown
in a flowchart (Figure 1) and discussed in detail in the next sections.
**2.1 Forward Radiative Transfer Model**
141        The first step in the training phase is to build a large data set of synthetic backscattered

Earthshine reflectance spectra from forward radiative transfer (RT) calculations. These
calculations are performed using the LInearized Discrete Ordinate Radiative Transfer (LIDORT)
model with the rotational Raman scattering (RRS) capability (Spurr et al., 2008). This version of
the model treats first-order inelastic Raman scattering in addition to all orders of elastic
(Rayleigh) scattering processes. Rotational Raman scattering occurs when a photon is scattered
at lower or higher energy levels than the incident radiation. RRS cannot be neglected; it is known
to be responsible for the Ring effect (Grainger and Ring 1962), which is a spectral interference
signature characterized by the filling-in of Fraunhofer lines and telluric-absorber features.
Allowing for RRS in the RT model leads to differences in calculated radiances compared to
those made with purely elastic scattering, as characterized by the filling-in factor. This quantity
is generally of the order of a few percent, consistent with estimates that 4% of the total scattering
in the atmospheric is inelastic (Young, 1981). Fundamentally the $SO_2$ layer height information
can be retrieved by backscattered radiance spectra because the amount of scattering occurring in
the overlying atmosphere is determined by the height of the volcanic $SO_2$ plume. This is
demonstrated by comparing two otherwise identical RT calculations with different $SO_2$ layer
heights (Figure 1a). At shorter wavelengths where Rayleigh scattering is stronger, there is less
backscattered radiance for the case with higher $SO_2$ plume height, particularly at shorter
wavelengths < 320 nm (Figure 1b). Likewise, the filling-in factor (Figure 1c) shows the
importance of including RRS in the RT calculations as in some cases there can be 2-3%
difference between the Raman and elastic calculations.



All LIDORT-RRS calculations in this study were performed for the 310-330 nm spectral
range, which captures strong $SO_2$ and ozone absorption features. The model is supplied with
ozone (Daumont et al., 1992) and $SO_2$ absorption (Bogumil et al., 2003) absorption cross
sections, atmospheric profile, ozone profile and a high resolution Fraunhofer solar irradiance
spectrum. The atmospheric profile has 48 layers and contains a temperature/pressure/height grid
from the standard US atmosphere, with an increased vertical resolution of 0.5 km below 12 km.
The ozone profile is determined by the total column amount, latitude zone and month as
specified in the TOMS V7 ozone profile climatology (Bhartia, 2002), while the $SO_2$ profile is
assumed to be a Gaussian shape with a full width half maximum (FWHM) of 2.5 km. The solar
spectrum is a re-gridded version of the high resolution synthetic solar reference spectrum
(Chance and Kurucz, 2010), originally with a spectral resolution of 0.01 nm. The re-gridded
version has a resolution of 0.05 nm, finer than that for OMI (0.16 nm sampling for a FWHM
spectral resolution of ~0.5 nm). The advantage of using this reference spectrum over the
instrument-measured irradiance is that only one set of calculations is needed; they can be applied
to multiple instruments and instrument cross track positions without utilizing unique measured
solar flux spectra for each situation. Using instrument-measured solar flux data can be more
accurate and better handles issues with instrument degradation. However, that would require the
inverse model to be re-trained whenever a new measured solar flux spectrum is used. Since we
expect the retrieval to be primarily sensitive to $SO_2$ absorption signatures, the radiative transfer
calculation was performed for a molecular atmosphere with no aerosol scattering.
In order to obtain a large number of different spectra, eight key physical parameters were
varied for the LRRS calculations. These parameters include solar zenith angle (SZA), relative
azimuth angle (RAA), viewing zenith angle (VZA), surface albedo, surface pressure, $O_3$ column
amount, $SO_2$ column amount and $SO_2$ layer height. The ranges of these parameters are given in
Table 1.
The number of calculations and the parameter sets for each simulation were determined through
a smart sampling technique (Loyola et al. 2016). A selective parameter grid with sets of
parameters for each simulation was established through the use of Halton sequences (Halton,
1962) in 8 dimensions. The calculations are continued until the moments of the output data,
mean and median converged across all wavelengths. In total around 200,000 calculations were
done to achieve sufficiently comprehensive sample size for the variation in the eight parameters





across all rows of OMI. This sampling was done in order to ensure that 1) each set of parameters
was unique and training data is diverse; and 2) that the sample size of the entire dataset is large
enough for the machine learning application.
**2.2 Data pre-processing**
After the RT calculations are completed, the spectra are convolved with OMI instrument slit
function. Since each cross-track position of OMI contains a unique slit function, the appropriate
function was applied based on the VZA input for that particular calculation. The VZA ranges
from 0-70° across all rows in the OMI swath, with the middle (nadir) rows having a VZA of
close to 0. For each row, only spectra within +/- 3° of the actual VZA were convolved with the
appropriate slit functions. In addition, Gaussian noise with a signal-to-noise ratio (SNR) of 1000
was added to the spectra. While the SNR of OMI tends to be lower (Schenkeveld et al., 2017),
adding too much noise can greatly decrease performance of the neural network (Table 3). At
SNRs of less than 500, the performance starts to increasingly degrade. Between 1000 and 500
SNR, there is an increase of around 0.1 km in RMSE. However, adding some degree of noise is
necessary to account for errors satellite instrument measurements.
Next, principal component analysis (PCA) was applied to the spectral dataset for each
row, in order to extract the most significant features of the spectra, and to reduce dimensionality.
Since each convolved sample consists of 142 wavelength points, the dimensionality of this
problem becomes very large. However, PCA transforms each sample to a set of weights based on
8 principal components (PCs). These principal components explain 99.998% of the variance in
the synthetic dataset (Figure 1A). Including additional PCs does not add any significant value to
the retrieval and may even lead to overfitting. Prior to starting the machine learning process, the
dataset is split into a training subset (90%) and a testing subset (10%). The training subset is used
for the neural network learning, while the testing subset only deployed verifying the performance
of the network to predict the output.
**2.3 Machine Learning using a Neural Network**
The 8 PCs, and selected parameters including the SZA, RAA, VZA, surface pressure and
surface albedo were used as input for training a MLPR, which is sometimes referred to as a deep
neural network. The output layer of the NN contains the effective $SO_2$ layer height. Column





amounts of $SO_2$ and $O_3$ were not included in the training or in the application stage because of
the large dependency of column amounts on $SO_2$ layer height and due to biases in OMI ozone
retrieval in the presence of the enhanced $SO_2$ plume, respectively. To improve stability, the
inputs (PC weights, SZA, VZA, etc.) and output (effective $SO_2$ height) are scaled between -0.9
and 0.9 according to the minimum and maximum of each input variable prior to input into the
NN. In a NN, the input and output layers are connected by hidden layers containing neurons
(also known as nodes). Each neuron is connected to others by a series of weights, by means of
which the input data is passed to the next level as a weighted sum of all inputs. The "tanh"
(hyperbolic tangent) activation function is applied at the hidden layers to further increase
stability in the NN. Inside the neural network, the Adam optimizer with a stochastic gradient
descent algorithm (Kingman et al., 2014) is used to minimize the loss function, in this case the
mean squared error (MSE) between the result of each iteration and the actual $SO_2$ layer height
used to generated the synthetic spectral sample. With each iteration, the partial derivative of the
MSE with respect to each node is calculated; this is used to update the weights. The training of a
NN progresses by cycling through iterations of the entire training dataset, called epochs, until the
training and validation MSE is minimized and there is no improvement to be obtained from
further training.
While there is a lot of flexibility in the setup of NN parameters, considerable trial and
error is needed to determine the best configuration that optimizes performance. The final
configuration of the NN in this study includes 2 hidden layers with 20 and 10 nodes in the first
and second layer, respectively. This was determined mostly through testing and analyzing the
performance of the NN with respect to both the synthetic test data set and real satellite
measurements. For this study, the training was done separately for each OMI row due to the
different VZAs and slit functions between rows; however, the configuration of the NN was kept
constant between rows. The only difference in the training is the number of training epochs
conducted for each row before the solution becomes optimal for that row. With this NN
configuration, the number of epochs was in the 200-300 range for all rows. The final trained
version of the NN, the inverse operator, contains the optimal weights needed to predict the $SO_2$
layer height from an input of separate test data.

**2.4 Application to satellite measurements**



In the application phase of the retrieval, the inverse operator is applied to OMI radiance
spectra, resulting in a predicted $SO_2$ layer height for each ground pixel in the OMI swath. For
this the OMI L1B Geolocated Earthshine radiance dataset is used. Since OMI only provides
absolute radiances, these data were normalized with respect to the same solar flux spectrum as
used in the generation of the synthetic spectra. In other words, the measured input becomes the
fraction of backscattered radiance to the incoming solar irradiance (i.e., reflectance spectrum).
Prior to normalizing, the irradiance spectrum was convolved with an OMI slit function for the
particular OMI row and orbit. The output is a predicted $SO_2$ layer height based on the input of a
radiance spectra and associated parameters, including VZA, SZA, RAA, surface albedo and
surface pressure, for a single OMI pixel. The irradiance spectrum is convolved with the
appropriate OMI slit function in order to have consistency in wavelength points between the
measured radiances, synthetic radiances and irradiance of each row. To follow the same
procedure as was used in the training step, the PCA operator from the training phase is applied to
the OMI spectra to perform the dimensionality reduction and obtain a set of PC weights for each
sample. The other inputs are VZA, SZA, RAA, albedo and surface pressure parameters from the
OMI data files. As in the training phase, all inputs are scaled to the [-0.9, 0.9] range. After $SO_2$
heights are retrieved separately for each row, one height value is given for each pixel (and
spectral sample). The application phase of the retrieval takes only 2-3 seconds for a given row.
This short duration includes the application of the training phase PCA operator to OMI
measurements, the scaling of inputs and the deployment of the inverse operator. The whole
process is repeated for each row in order to get a prediction for an entire OMI swath. For some
rows the retrieval is unreliable due to the row anomaly, which negatively affects the quality of
the OMI L1B radiance data at all wavelengths and consequently L2 retrievals.

**3 Impacts of various parameters on the performance of the trained inverse model**

From the training phase, it becomes clear that the performance of the algorithm will
depend on several factors. As demonstrated in Fig. 3, an important factor is the $SO_2$ column
amount. Overall, the NN makes better predictions for the test data subset for $SO_2$ amounts > 40
DU. Below 40 DU, information content on the layer height to be retrieved becomes increasingly
small, as evidenced by large differences between predicted heights and those in the actual test set
(Figure 3a). Additionally, larger $SO_2$ loadings result in greater sensitivity between two heights,


as seen by comparisons of $SO_2$ height Jacobians for multiple amounts (Figure 2A).
Quantitatively, if samples with $SO_2$ amounts less than 40 DU are excluded, the RMSE decreases
from 1.48 to 1.15 km (Table 2). We can therefore expect the retrieval to produce reasonable
results for larger volcanic eruptions. In widely dispersed plumes where the $SO_2$ VCD is low, the
retrieval would be biased and less useful. The second major dependency is on SZA. The problem
here stems from the occurrence of relatively large errors in RT modeling due to shallow light
paths and lower OMI SNR at the higher SZAs. Reasonably accurate results are to be expected
only for SZA < 75°. Figure 2b shows significant differences in predicted and actual heights in
spectra associated with large SZAs, after removal of low VCD samples. For the final training
approach, it was therefore necessary to exclude spectra with large SZAs. Dependencies on other
physical parameters are small when compared with these two issues discussed here, although
there is some evidence that high surface albedo also increases error. If we remove spectra with
albedo > 0.6 there is a minor improvement in RMSE from 0.93 to ~0.89 km. However, even with
strong volcanic $SO_2$ signals, we can realistically expect that on average the absolute error to be at
least 1 km, due to inherent simplifications in the neural network retrieval approach. The errors in
actual retrievals using OMI data are expected to be larger (see Section 4.4).

**4. OMI $SO_2$ Effective Layer Height Results**

308       For testing the FP_ILM retrieval on OMI data, four volcanic eruption cases with sufficiently

strong $SO_2$ signals were selected (i.e. where peak $SO_2$ VCDs were greater than 40 DU). Each
case is described in detail in the following subsections. For each case, comparisons were made to
other satellite-derived datasets where available, for example the CALIOP lidar onboard
CALIPSO, the IASI $SO_2$ layer height retrieval (Clarisse et al., 2014), and the GOME-2
(Efremenko et al., 2017) and TROPOMI retrievals (Hedelt et al., 2019). It is important to note
that the CALIOP lidar only indicates the height of the ash plume and not the $SO_2$ height.
Although ash and $SO_2$ plumes are often collocated, this is not always the case, making direct
comparisons difficult.

**4.1 Kasatochi (2008)**
Kasatochi is a volcano located on the Aleutian Islands of Alaska (52.178°N,175.508°W). It
underwent a series of eruptions beginning late in the day on August 7th, 2008, which injected





great amounts of ash and $SO_2$ into the stratosphere. Overall the explosion released roughly 2
million tons of $SO_2$, at the time the highest $SO_2$ loading since the Mt Pinatubo eruption (Yang et
al, 2010). $SO_2$ effective layer heights retrieved using the machine learning model for OMI (orbit
21650) on August 10th, 2008, were around 11-12 km with some portions being slightly lower
(Figure 4a). This is in reasonable agreement with previous $SO_2$ height retrievals of 9-11 km
which used the ISF algorithm for OMI (Yang et al., 2010). Likewise, Nowlan et al. (2011)
showed that the majority of the plume was around 10 km, and up to 15 km in some parts.
Furthermore, there is agreement with IASI (Figure 4b) and CALIOP data (Figure 4d) which
showed plume heights of 10-12 km and 12.5 km respectively. It is important to note that the
IASI overpass occurred later in the day than those for OMI and CALIPSO. Another verification
source we used was the GOME-2 $SO_2$ layer height retrieval that uses FP_ILM (Efremenko et al.,
2017). The study found a height of around 10 km and up to 14 km in areas of high $SO_2$ loading
for August 10th (Figure 4c). Although the OMI results agree well in general with the results of
these studies and datasets, the retrieval is less sensitive with respect to detecting variability in the
$SO_2$ layer height within the plume, compared to the GOME-2 case. It should be noted that the
GOME-2 overpass occurred earlier in the day than OMI.
**4.2 Kelud (2014)**
Kelud is a stratovolcano located in East Java, Indonesia (7.935°S, 112.315°E). It erupted on
February 13th, 2014 at 1550 UTC, in the process depositing ash in a 500 km diameter around the
volcano and leading to mass evacuations from nearby towns. The OMI retrieval results indicate
that the maximum height of the main plume was 18-19 km (Figure 5a), although other studies
suggest that several small layers of $SO_2$ and ash were located as high as 26 km (Vernier et al.,
2016) on the previous day. However, the $SO_2$ loading at that level was most likely too low for an
accurate retrieval using OMI radiances. CALIOP lidar detected ash plumes at around 19.5 km
and the IASI retrievals registered the plume at 17.5 km over the same area as that for OMI. The
height of the ash plume from this eruption was also estimated using Multifunctional Transport
Satellite (MTSAT 2) observations and transport modeling (Kristiansen et al., 2015). That study
found an injected height of around 17 km, which is in agreement with the OMI result, especially
when considering the PDF of the heights (Figure 6b). We note here that only a small portion of
the plume was retrieved with our algorithm, given the relatively low $SO_2$ VCDs and interference





due to the OMI row anomaly. It is promising to note that the OMI retrieval was able to identify
heights at the upper end of the height range used in the training phase. On the other hand, while
the retrieval can extrapolate to heights above 20 km, the accuracy would likely degrade due to
the lack of training data with heights outside of this limit.

**357    4.3 Calbuco (2015)**


The Calbuco eruption in April 2015 and the Kelud eruption in February 2014 are both significant
volcanic events that injected $SO_2$ plumes well above 10 km into the atmosphere. We have chosen
to apply the FP_ILM to these events even though they have somewhat lower $SO_2$ VCDs as
compared with those from Raikoke and Kasatochi; nevertheless, peak $SO_2$ columns  with~60-70
DU should allow reasonable accuracy in our retrievals (see section 2).

364          The Calbuco volcano is located in Chile (41.331°S, 72.609°W). The primary eruption

had a volcanic explosivity index (VEI) of 4 and occurred on April 22nd with little warning. The
primary plume ascended higher than 15 km, while plumes from smaller subsequent eruptions
stayed in the troposphere. The volcanic plume spread northeast in the following days, resulting in
flight cancellations at Uruguayan and south Brazilian airports. The OMI-retrieved $SO_2$ effective
layer heights in the area of greatest VCD was in the 15-17 km range. In the same region, IASI
results (Figure 5c) show similar plume heights, approximately around 15 km, although as with
the previous events, the overpass times of the two instruments are different. CALIOP lidar shows
the ash plume to at roughly 17 km (Figure 5e). Unfortunately, the overpass of CALIPSO occurs
over an area of OMI's swath that is affected by the row anomaly, and this makes a direct
comparison unfeasible. Nevertheless, the CALIPSO aerosol layer height is still comparable to
OMI-retrieved effective $SO_2$ layer heights for the portion of the plume further to the west. The
retrieval for OMI is consistent with the other instruments for $SO_2$ plumes, with the exception of
that part of the plume with $SO_2$ below 30-40 DU (see Figure 3A), for which results were not
plotted in Figure 5a due to lower biases.

**380    4.4 Raikoke (2019)**

The eruption of the Raikoke stratovolcano (48.2932°N, 153.254°E), located on the Kuril Islands
of Russia, occurred on June 21st, 2019 at 1800 UTC. A series of explosions during the eruption
sent large amounts of ash and $SO_2$ into the lower stratosphere. Maximal loadings of $SO_2$





measured by OMI and other sensors exceeded 500 DU. In the following days the plume
underwent dispersion and spread out over the northern Pacific Ocean and later over eastern
Russia. Early estimates of plume injection height for the eruption were predominantly in the 10-
13 km range with potentially larger heights in some areas of the plume. In Figures 7a and 7b, the
$SO_2$ effective layer heights retrieved from OMI data are shown for the Raikoke plume on June
23$^{rd}$ and June 24$^{th}$ respectively. The plume heights for both days are predominantly in the range
10-12 km, although some areas of the plume had estimated peak heights of 13-14 km. In
comparison, the TROPOMI results show slightly larger heights (13-14 km) for June 24$^{th}$ and
similar heights to OMI for June 23$^{rd}$ (Figure 7c and 7d). The IASI $SO_2$ height product also shows
fairly good agreement, with heights mainly at the 10-11 km level (Figure 7e and f). It is also
useful to look at a distribution of heights predicted for the domain (Figure 8) in order to get a
more quantitative comparison between the datasets. Based on this distribution, there is clearly a
1-2 km difference between the most probable heights from OMI and those from TROPOMI for
June 24$^{th}$ (Figure 8b and 8d) and slightly lower heights in the distribution for IASI. Note that
points with lower than 30 DU are not included in the PDFs for all sensors. The results are also
compared with CALIOP lidar onboard CALIPSO, which shows ash plume heights of 12-13 km
for both days (Figure 9a and 9b). Although there is overestimation for some OMI pixels,
especially for June 24$^{th}$, the section of the plume with the CALIPSO flyover has similar heights
(around 12.5 km) to  lidar-determined aerosol layer altitudes. Lastly, we note that a recent study
highlighted probabilistic height retrievals using the Crosstrack Infrared Sounder (CrIS) for
Raikoke. This study found a median height of 10-12 km across a large part of the plume,
however with some areas upwards of 15 km. While there are some notable differences across all
of the datasets, the OMI retrieval for this case falls within the general consensus of plume height
estimates  for this volcanic event.
**4.5 Discussion of errors**
It is clear that predicting $SO_2$ layer height with FP_ILM is an efficient process, but one that is not
flawless in terms of accuracy. As comparisons between instruments have showed, on average
there were 1-2 km differences in heights, especially for the Raikoke event, although we consider
this to be good agreement given the estimated RMSE associated with this retrieval. In this
regard, the height retrieval is more likely to give a rough estimate of the $SO_2$ plume height rather



than a precise prediction. Comparison errors result from differences in instruments, forward
model assumptions and retrieval techniques. For instance, IASI is an IR-based instrument and its
retrieval does not use FP_ILM. Therefore exact agreement with IASI results is difficult to
achieve, although its retrievals serve as a good verification dataset. For OMI and TROPOMI,
which both use UV spectra and an FP_ILM algorithm approach, there are instrument differences
such as the pixel size, noise, radiometric accuracy and the level of degradation. TROPOMI has a
much finer spatial resolution compared to OMI, with footprints typically 5.5x3.5 $km^2$ up to
maximum size 7x3.5 $km^2$ ; TROPOMI also has as enhanced $SO_2$ signals. Consequently,
TROPOMI is better able to resolve localized variations in the height throughout the plume, and
is likely to be more accurate overall. OMI retrievals show more or less uniform height levels
across the entire plume with the peak heights in areas with the best $SO_2$ signal. However, current
TROPOMI L1 data are known to have issues with instrument degradation and radiometric
accuracy in the UV spectral range; this could be a potential contributing factor to explain the
differences between the two instruments.. It is also worth mentioning that CALIOP lidar profiles
sometimes show disagreements with OMI retrieved heights, because CALIOP only identifies the
height of the ash or aerosol plume. It also offers a comparison for only a single cross section of
the entire plume per orbit. Overall, the consensus provided by different instrumental datasets can
provide a reasonable estimate for the $SO_2$ layer height, and if done in near real time, can aid in
decision making with regards to aviation safety.
Another source of error is present in the training phase. One difficulty here is finding the ideal
choice of neural network setup. With many parameters to consider, such as the number of input
PCs, number of layers, number of nodes, learning rate, regularization, weight initialization, etc.,
it is very time consuming to optimize the neural network setup. We have found a relatively
simply configuration that performed reasonably well with both test data and real OMI
measurements for all scenarios and events considered. It is difficult to improve results further
than ~1 km absolute error, even in the training phase. In the application phase, additional error
comes from the differences between synthetic spectra and real satellite measurements with noise
errors. For example, with an SNR of 500 used in training, which is a typical noise level for OMI,
the RMSE of the neural network prediction is around 1.25 km (Table 3). This can be considered
the lower limit of retrieval error when the inverse operator is used on OMI measurements.
Lastly, some deviations between the measured and synthetic training spectra originate from the




RT modeling. The calculations contain several assumptions including the $SO_2$ plume shape,
atmospheric profiles, gas profiles, and a molecular scattering atmosphere. Further testing is
required in order to determine if the inclusion of aerosols in RT calculations would improve the
algorithm performance.
**5 Conclusion**
In this study we have introduced a new algorithm for OMI retrievals of the volcanic $SO_2$
effective layer height from UV earthshine radiances. This algorithm is based on an existing
FP_ILM method which combines a computationally time-consuming training phase with full
radiative transfer model simulations and a machine learning approach to develop a fast inverse
model for the extraction of plume height information from radiance spectra. Fast performance
means that the algorithm can be considered for operational deployment, given that the retrieval
of a $SO_2$ layer height prediction from the inverse model takes only a matter of milliseconds for a
single OMI ground pixel. For the training, a synthetic dataset of earthshine radiance spectra were
created with the LIDORT-RRS RT model for a variety of conditions based on choices of 8
physical parameters determined with smart sampling techniques. A dimensionality reduction was
performed through PCA in order to reduce the complexity of the problem and to separate those
features that best capture the great majority of variance of the dataset; 8 principal components
were sufficient for this purpose. Dimensionally-reduced data together with the associated
parameters were used to train a double hidden-layer neural network to predict $SO_2$ plume height
from any given input data. The  PCA from the training phase and the inverse operator resulting
from the optimal NN framework were then applied to real satellite radiance spectra and
parameters to get  retrieved values of $SO_2$ plume heights for several volcanic eruption events.

470         Through comparisons with CALIPSO lidar overpasses, TROPOMI and IASI retrievals, it

was shown that the retrieval for OMI can estimate reasonable $SO_2$ layer height for all the events
considered, with absolute errors of up to 1.5 km. These results can give an indication on
approximate plume heights achieved during medium- to large-scale eruptions, which can lead to
important decisions in aviation hazard mitigation. For all events treated in this study, there was
general agreement with CALIOP lidar, although locations of the CALIPSO flight path for the
Kelud and Calbuco cases were unable to be retrieved due to OMI row anomaly issues.



Uncertainties and error sources in using this approach which open up possibilities for
future work in improving the accuracy and robustness of this method. One assumption that was
made is that ash and sulfur dioxide plumes are mostly collocated when using CALIPSO as a
source to verify the plume height. Although this is often true, dispersion of the plume in the days
following the eruption can separate the two components. Therefore, tracking these plumes
become challenging when using reflectance spectra alone; further analysis also may need to
include trajectories or wind data. Secondly, the model was trained on synthetic spectra calculated
for molecular atmosphere conditions in the absence of any aerosol loading. The impact of
including aerosols in the simulations is another subject for a follow-up study. We also intend to
generate data sets of synthetic spectra by using a vector RRS model to account for polarization.
Other future work will include extending the application of FP_ILM to the Suomi-NPP OMPS
instrument as well as exploring the ability to predict multiple outputs at once from this approach.

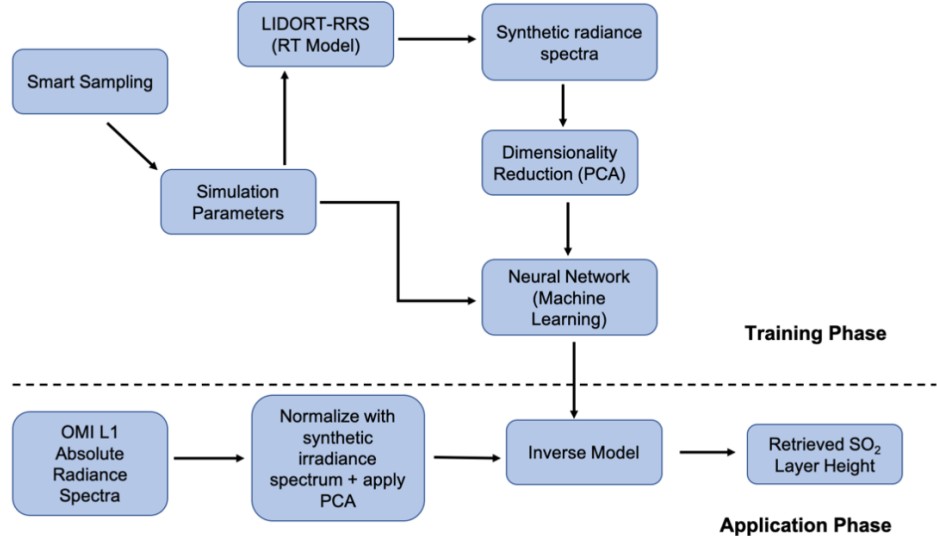


**Figure 1:** The flowchart of the FP_ILM methodology for retrieving OMI SO$_2$ Effective Layer Height.
The steps above the dashed line are part of the training phase which is done prior to incorporation of OMI
measurements. The application phase involves deployment of the trained model to the OMI radiance
measurements to obtain estimates of effective volcanic SO$_2$ layer heights.

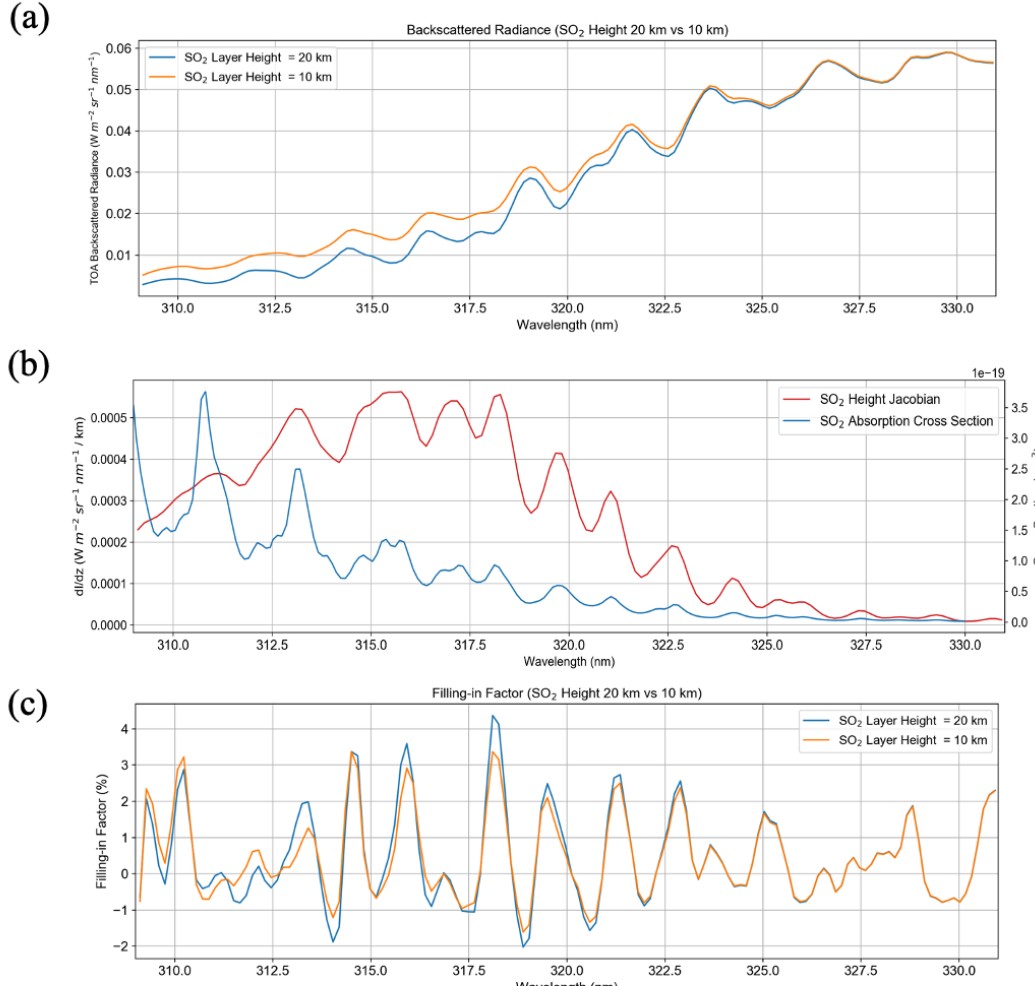

**Figure 2:** (a) Simulated top of the atmosphere (TOA) Earthshine radiances for two different $SO_2$ layer heights (10 km and 20 km) from the LIDORT-RRS model. Also shown:(b) the $SO_2$ height Jacobian (change in radiance per km between the two spectra) along with the absorption cross-sections of $SO_2$ for reference; (c) the filling-in factor. The filling-in factor is defined as the difference between the total and elastic-only radiance results, divided by the total radiance, expressed as a percentage. An $SO_2$ column amount of 200 DU was used in the two calculations and all other parameters were kept constant except for the $SO_2$ layer height.



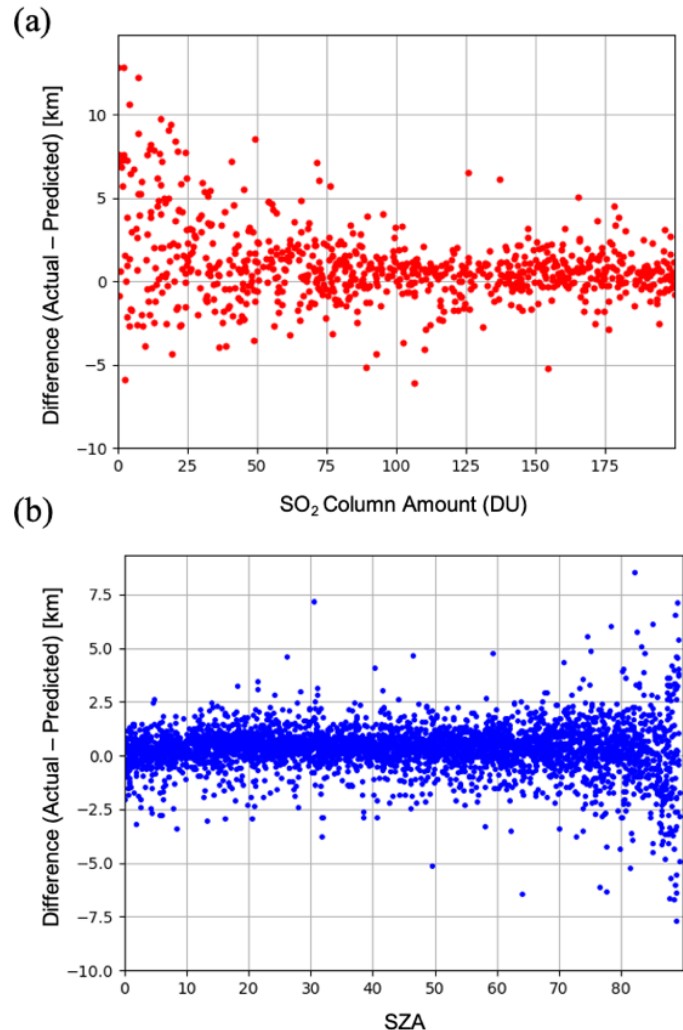

**Figure 3:** Dependence of retrieval errors on (a) SO$_2$ amount and (b) SZA for cases with SO$_2$ VCD > 40
DU. The error is defined as the difference between the SO$_2$ layer height predicted by the neural network
using inputs from the independent test set, and the actual height from the same samples. The test set
comprises 10% of the original spectral dataset withheld from training the neural network. The plots show
that the retrieval error is mostly within +/- 2.5 km for SZA < 70, but increases significantly for large
SZAs.

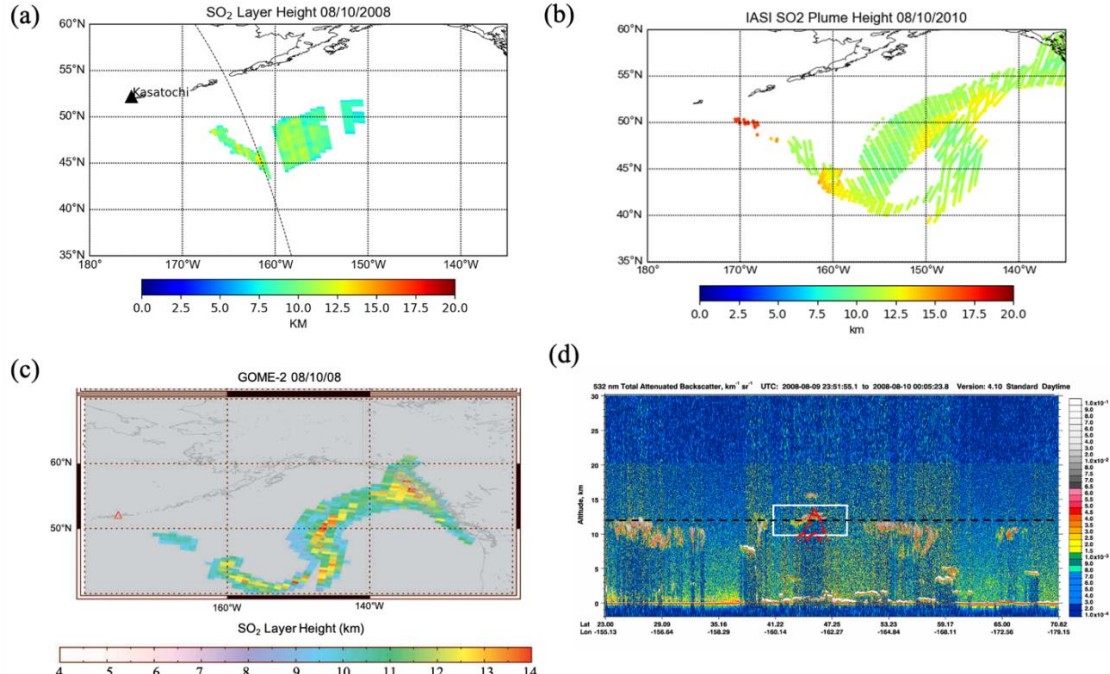

**Figure 4:** Comparison between the volcanic plume heights from (a) OMI, (b) IASI, (c) GOME-2 and (d) CALIOP lidar 532-nm attenuated backscatter, for the 2008 Kasatochi eruption . The white rectangle in (d) shows the area of the volcanic plume on the vertical profile. The GOME-2 retrieval figure was obtained from Efremenko et. al 2017. The black dashed line in (a) shows the CALIPSO track. Some rows of OMI in this case were affected by the row anomaly, as seen by the gaps in the plume. The red dots in (d) show the OMI retrieval near the CALIPSO path.

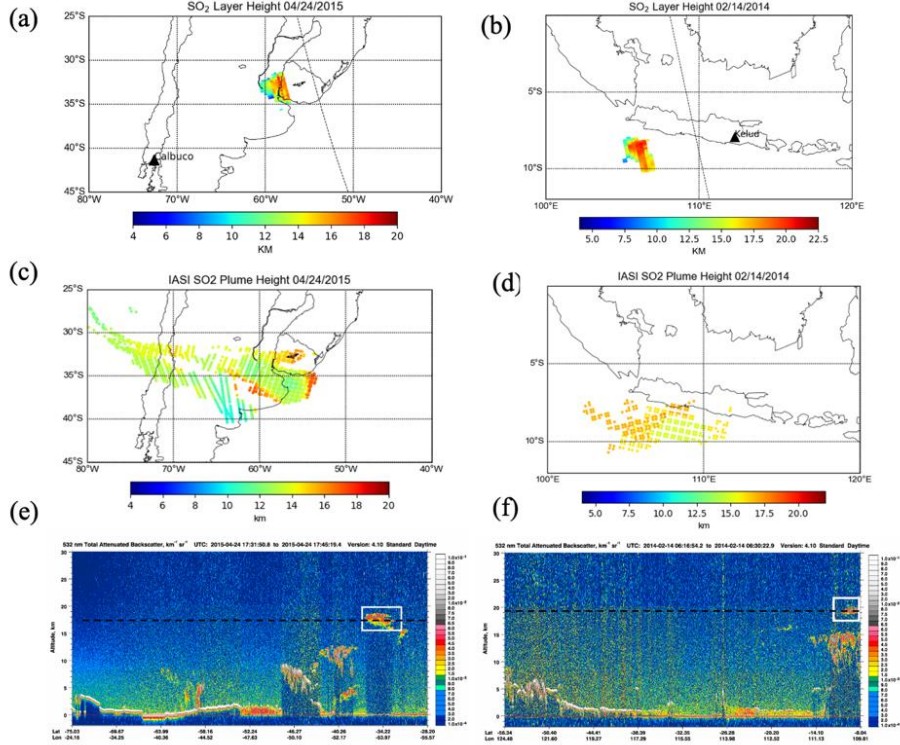

**Figure 5:** Comparisons of plume heights for the 2015 Calbuco eruption (left) and the Kelud eruption (right) for OMI (a,b), IASI (c,d) and 532-nm total attenuated backscatter from the CALIOP lidar (e,f). For OMI, only pixels with > 30 DU of $SO_2$ are shown and retrievals were unavailable for some parts of the plume due to the row anomaly. The black dashed line in (a) and (b) marks the CALIPSO track. The white rectangles in (e) and (f) show the location of the plume in the lidar profile. Unfortunately, direct comparison with CALIPSO is not possible due to obstruction by the row anomaly

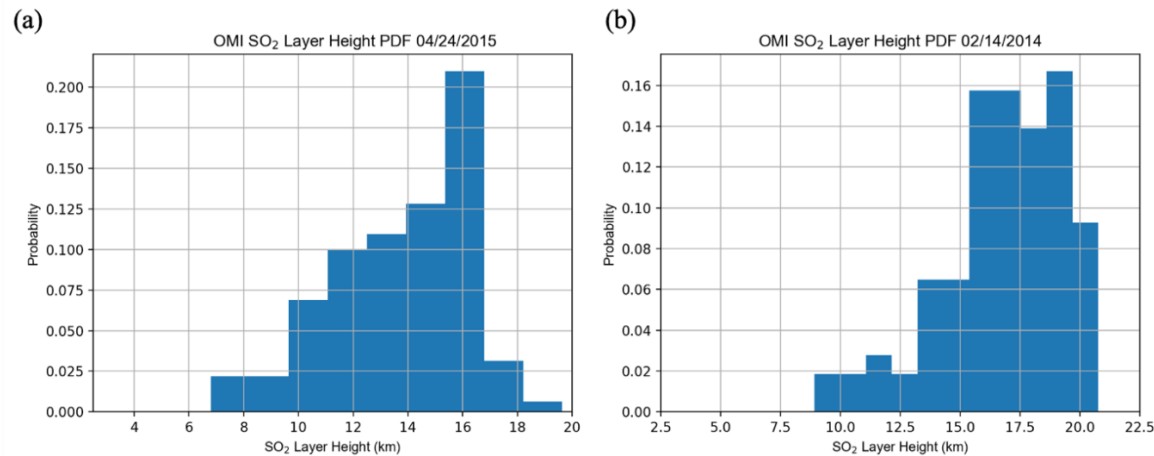

**Figure 6:** Probability histograms of SO₂ effective layer height retrievals for (a) the Calbuco
eruption on April 24, 2015 and (b) the Kelud eruption on February 14, 2014.



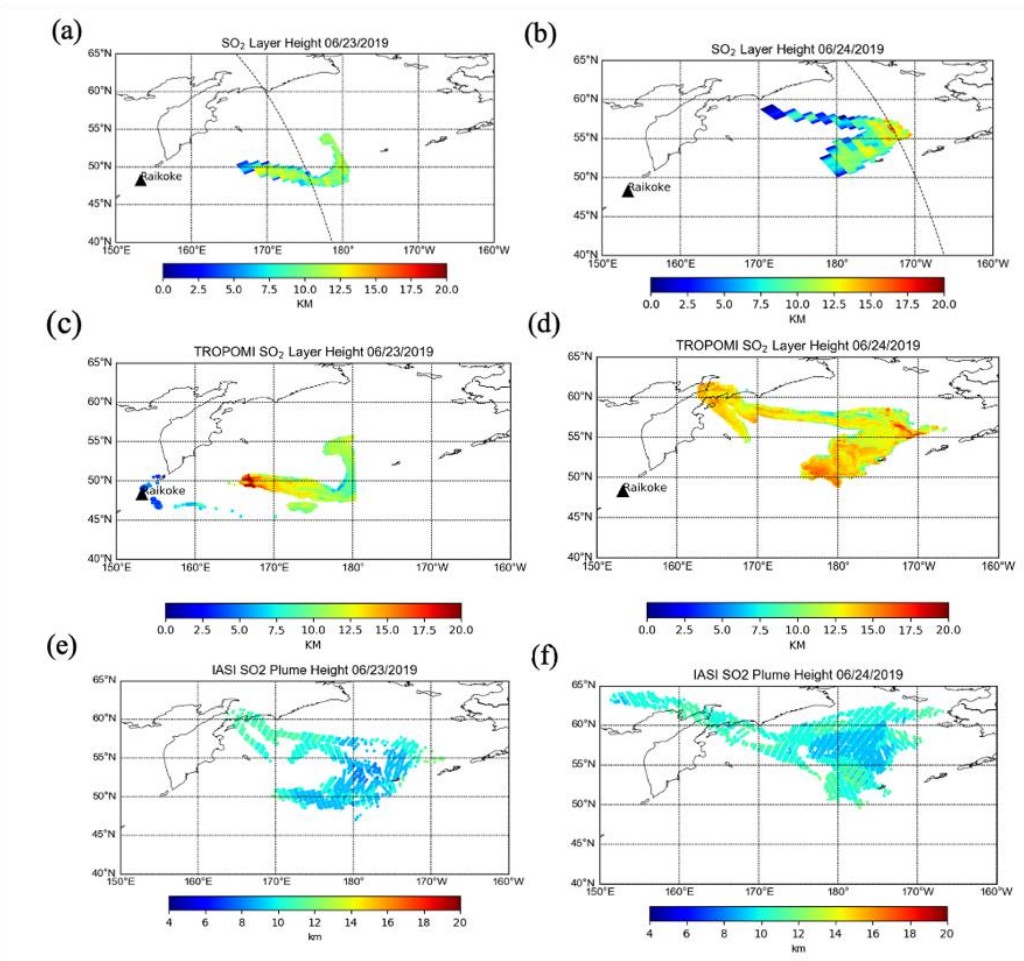

**Figure 7:** The SO₂ layer height retrieval for the Raikoke eruption plume on June 23[rd], 2019 (left) and June 24[th], 2019 (right) for the OMI (a, b), TROPOMI (c, d) and IASI (e, f) instruments. For all 3 sensors, only pixels where SO₂ VCD > 30 DU are shown. Note that for IASI, the color scale has been changed slightly in order to make differences within the plume more visible.

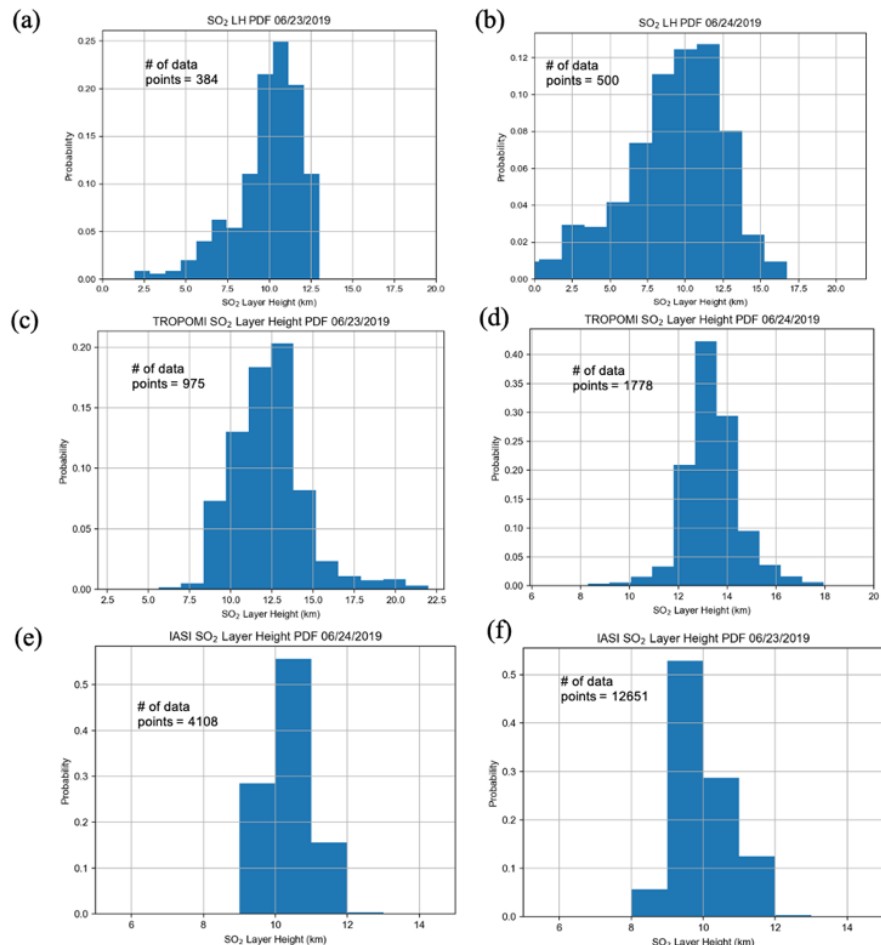

**Figure 8:** Probability histograms of SO₂ layer height retrievals for (a,b) OMI and (c,d),
TROPOMI on June 23rd, 2019 (left) and June 24th, 2019 (right) and (e,f) IASI. Only pixels with
SO₂ column amount greater than 30 DU are included. These plots correspond to the results
plotted in Figures 4a-f.



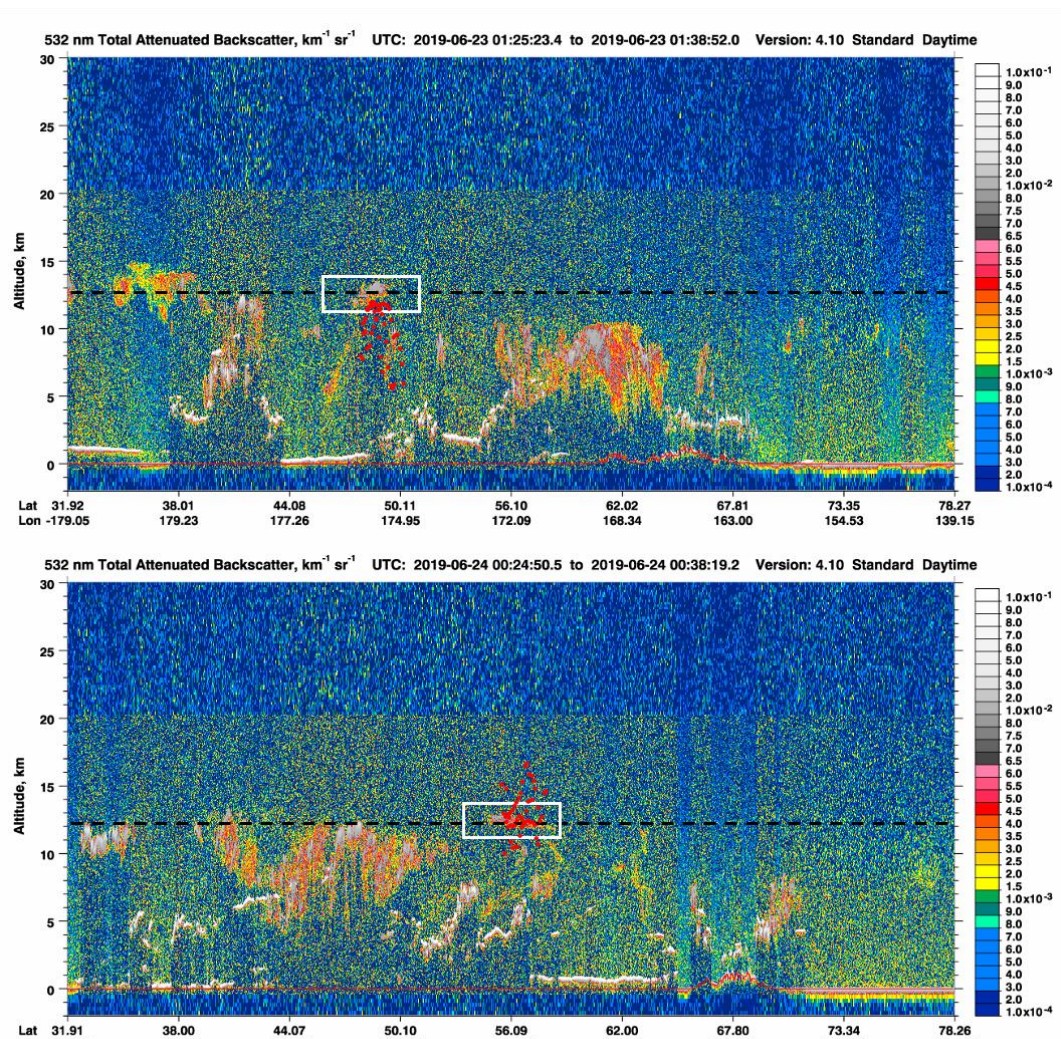

**Figure 9:** CALIOP lidar 532-nm attenuated backscatter for the Raikoke eruption on (a) June 23[rd] and (b) June 24[th], 2019.The while rectangle denotes the volcanic plume signature, with the black dashed line symbolizing the height. Red dots show the results from the OMI retrieval along CALIPSO's flight path. The flyover occurred shortly after 00:30 UTC, around the same time as OMI.



**Table 1:** Ranges of the eight physical parameters varied in LIDORT-RRS for the synthetic
spectra calculations.

| Parameter | Range |
|---|---|
| Solar Zenith Angle | 0-90° |
| Viewing Zenith Angle | 0-70° |
| Relative Azimuth Angle | 0-180° |
| Surface albedo | 0-1 |
| Surface pressure | 250-1013.25 hPa |
| $O_3$ VCD | 225-525 DU |
| $SO_2$ VCD | 0-1000 DU |
| $SO_2$ Layer Height | 2.5-20 km |



**Table 2:** The RMSE and the mean absolute difference of all data points in the test set under
different conditions. For each condition, the appropriate points were removed and not included in
calculating the errors. All cases in this table used synthetic training spectra with added SNR
564 1000.

| | All cases | $SO_2 > 20$ DU | $SO_2 > 40$ DU | $SO2 > 60$ DU | SZA < 75º | $SO_2 > 40$ DU and SZA < 75º | Albedo < 0.6 | $SO_2 > 40$ DU , SZA < 75º , Albedo < 0.6 |
|---|---|---|---|---|---|---|---|---|
| RMSE | 1.487 | 1.216 | 1.150 | 1.109 | 1.281 | 0.931 | 1.524 | 0.895 |
| Absolute Mean Difference (km) (Predicted – Actual) | 0.910 | 0.834 | 0.803 | 0.782 | 0.795 | 0.697 | 0.895 | 0.667 |


**Table 3:** The RMSE and the mean absolute difference of all data points in the independent test
set after adding noise as indicated by different SNR values. All other parameters and input data
were kept constant. SZA < 75 degrees and SO2 VCD > 40 DU were excluded from the test set
for these comparisons.

| | No noise | SNR=1000 | 750 | 500 | 200 | 100 |
|---|---|---|---|---|---|---|
| Mean Absolute Difference (y_known - y_pred) (km) | 0.6805 | 0.697 | 0.7265 | 0.7773 | 0.8859 | 1.1825 |
| RMSE (km) | 1.093 | 1.150 | 1.176 | 1.2514 | 1.513 | 1.9 |
| R-coefficient | 0.989 | 0.985 | 0.984 | 0.981 | 0.973 | 0.957 |


**Appendix A: Supplemental Figures**

**Figure 1A:** Explained variance ratio as a function of the number of principal components of the
spectral dataset.

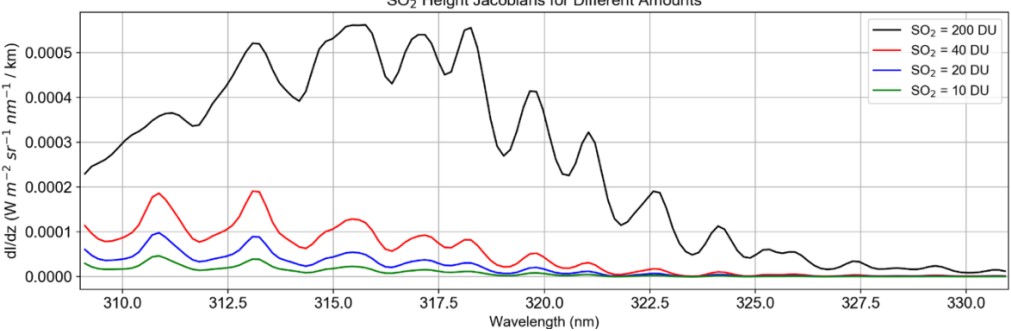

**Figure 2A:** $SO_2$ Height Jacobians (dI/dz) for 4 different assumed $SO_2$ column amounts. The
Jacobians were calculated from the difference between two radiance spectra with 10 km and 20
km $SO_2$ height. All other physical parameters were identical in the calculation of the spectra.

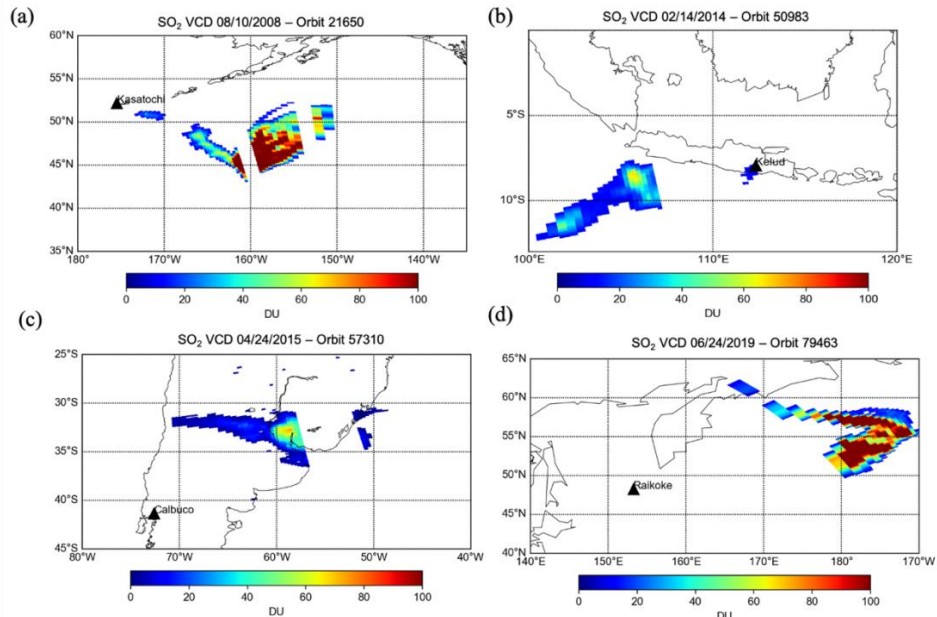

**Figure 3A:** OMI $SO_2$ VCD for the four volcanic cases: (a) Kasatochi on August 10[th], 2008, (b) Kelud on February 14[th], 2014, (c) Calbuco on April 24[th], 2015 and (d) Raikoke on June 24[th], 2019. In these maps, only pixels with $SO_2 > 10$ DU are shown.

**Data availability**. OMI SO2 L1 and L2 data can be accessed via the Goddard Earth Sciences Data and Information Services Center (GES DISC) at https://earthdata.nasa.gov/eosdis/daacs/gesdisc. IASI SO2 LH data is available via the IASI AERIS portal https://iasi.aeris-data.fr/. NASA CALIPSO data can be downloaded from https://www-calipso.larc.nasa.gov/ and images can be found at https://www-calipso.larc.nasa.gov/products/lidar/browse_images/production. TROPOMI L2 SO2 data can be obtained at https://s5phub.copernicus.eu/dhus/\#/home while the LH is experimental and is not yet publicly available online. The results of OMI SO2 layer height retrieval presented in this study can be obtained from the author by request.

**Author contributions**. NF wrote the manuscript and performed most computational and model work in this study. The project was conceived and overseen by CL and NK. DL and PH provided the TROPOMI SO2 LH retrieval and input on the comparisons in the paper. PH also offered support relating to the machine learning aspect of the study. RS is the original developer of the LIDORT-RRS code and provided related support, as well as input to the relevant sections of the manuscript. RD is an advisor of NF and provided additional input to the paper and was involved in project planning.





**Competing interests**. The authors declare that they have no conflict of interest.

**Acknowledgements**. We would like to acknowledge the NASA Earth Science Division (ESD) Aura Science
Team program for funding of the OMI SO$_2$ product development and analysis (Grant # 80NSSC17K0240).
OMI is a Dutch/Finish contribution to the NASA Aura mission. The OMI project is managed by the Royal
Meteorological Institute of the Netherlands (KNMI) and the Netherlands Space Agency (NSO).

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
