# Peer review of "Machine Learning Approach"

_Atmospheric Measurement Techniques, 2020_

## Referee Comment (RC1) · Anonymous Referee #1 · 13 Nov 2020

I focus my review on the portion of the paper dealing with deep neural networks (DNNs) and data processing. Despite some analysis choices made that I cannot understand, the ideal of using neural networks to retrieve SO2 plume heights is interesting. I am disappointed that the evaluation of the performance of the DNNs is not well presented. A series of sensitivity experiments could be conducted to improve the quality of this study.

**Comments**

Line 194-195: How large is the sample size large enough? I think it would be good to have the number of trainable parameters in the DNNs reported here.

Line 244-246: Have you tried to increase the number of neurons in each layer of the

[Figure]

DNNs or increase the number of hidden layers in the architecture? A sensitivity analysis during a "fine-tuning" process is very helpful for the optimization of DNNs.

Line 233-234: The tanh activation function is more frequently used in classification problems. For regressions, the rectified linear unit (ReLU) and parametric ReLU (PReLU) are more used.

Line 217: How do you control overfitting? Do you have a validation set? A validation set should be a small subset sampled randomly from the training set, over which the performance of the DNNs is checked after each epoch of the training. Sometimes the training loss gets reduced, but validation loss is not.

Figure 3: Relative errors are much more meaningful here because the "truth" of volcanic SO2 layer heights is not a constant set. It is also clear that the predictability of volcanic SO2 layer heights has a strong correlation with SO2 column information. Since OMI is biased, have you tried to add SO2/O3 column information from other sources as predictors?

Line 229-231: If you scaled the parameters to be within the max/min range, plus the tanh activation function you used, the gradients of DNNs with respect to trainable parameters in DNNs would not be sensitive to predictors closer to the max/min values. The degraded performance of your DNNs shown in Figure 3b could be a result of this. Maybe try batch normalization?

**Specific points**

- Regarding the stability of DNNs, you could also consider to add skip connections. This is not technically hard. It would smooth the surface of the loss function and reduce the number of local minima. (arXiv:1712.09913)

- It is difficult to evaluate the performance of your DNNs by comparisons shown in Figures 4, 5, 7 and 9. A heat map could be very helpful here, between the predicted volcanic SO2 layer heights and those retrieved from other satellites.
- Table 2 and 3 show statistics intercompared within the synthetic data set. I think it is meaningful to also have statistics compared with other independent satellite retrievals.

- If you do have access to other retrievals of volcanic SO2 layer heights, then I would suggest a multi-stage training. In stage 1, synthetic data sets are used. In state 2, you can keep training the model from stage 1, using a subset of real retrievals as the training set and the other as the testing set. If only synthetic data sets are used and the forward model generating these sets is not perfect, then the trained model must have a degraded performance compared to the forward model (due to errors from deep learning model itself). Moreover, during the forward model calculation, if the column data (O3 and SO2) are sampled within some range, then your DNNs would have a difficult time in predicting outliers and extrapolation should be very careful. Speed would be the only advantage then. The data-driven nature of DNNs should be taken advantage of.

- Bayesian neural networks are ideal for such prediction problem, when there are uncertainties associated with the outputs. This could be future work due to the technical complexity. But you could still alter the random seed to generate an ensemble of DNNs, such that you can provide a rough estimate of uncertainties on the predictions.

**Minor comments**

Line 61: replace VCD by vertical column density (VCD)

Fig 4: can you change the colorbar of (c) to the same scale as (a) and (b)?

[Figure]

---

## Referee Comment (RC2) · Anonymous Referee #2 · 17 Nov 2020

This paper describes the implementation of an existing machine learning algorithm to retrieve volcanic SO2 layer height from the OMI satellite instrument. The paper details initial promising results, and I would recommend for publication.

The paper is based on an existing algorithm (FP_ILM) which has been applied for SO2 layer height retrievals in other instruments (particularly Efremenko et al, 2017 and Hedelt, 2019.). While these are clearly reference in the introduction, it would be good to make clearer in subsequent sections (particularly Section 2: Methodology), which sections directly follow the earlier papers approach and reuse existing inputs, and which parts are implemented specifically for OM

It is also recommended that the authors address the following specific points:

[Figure]

Specific Comments:

Abstract, line 34-35: The last sentence of the abstract states "This approach offers a promising prospect of using physics-based machine learning applications to other instruments." However, as this algorithm did not originate in this study or for the OMI instrument, and so has already been demonstrated for a number of other instruments, as phrased here the statement doesn't seem valid. I would suggest the sentence is either rephrased (if that wasn't it's intended meaning), or removed.

Introduction, lines 99-104: The section describes IASI retrievals of SO2 height, and follows it with the phrase ('For these techniques, extensive radiative transfer modeling is needed...). However, in the Clarisse 2014 paper referenced, this describes a fast retrieval scheme, where this statement may not follow. I would suggest checking and amending the text appropriately.

Section 2.2: In the first paragraph, there is a discussion on the impact of the SNR used, with reference to Table 3. First, I would suggest moving some of this discussion to Section 3, where the other performance impacts are discussed, as concepts used here, such as the split between training and test data are not explained until later in section 2. Secondly, it wasn't clear to me what Table 3 represents. Is this the impact of applying the same varying SNR's to both the training set and test set of data, or is it the impact of applying a SNR of 1000 to the training set and different SNR's to the test set? (To me the text reads one way and the caption the other). If it's the first case, have the authors also looked at the impact of applying the data trained with their chosen (better than reality) SNR to synthetic data which has had the realistic OMI noise applied? When the inversion is applied to real data, this will be the SNR level that the algorithm has to cope with, so would be a valuable indicator of expected performance.

Section 2.4 Line 264-266: "The output is a predicted SO2 layer height based on the input of a radiance spectra and associated parameters, including VZA, SZA, RAA, surface albedo and surface pressure, for a single OMI pixel." Is this sentence in the

right place – I found the flow of the paragraph a bit confusing, as it then jumps back to talking about convolving the irradiance spectra and then applying PCA?

Section 2.4, Line 278: The text assumes that readers will be familiar with the OMI row anomaly, which may not be the case – it would be useful to explain this somewhere.

Section 3: Tables 2 and 3 need more explanation in the text here e.g. RMSE is mentioned, but it's not explicitly stated what this represents anywhere. Also, from their captions, I would have expected the RMSE numbers in Table 2 for VCD > 40 and SZA < 75, to be the same as the RMSE in Table 3 for SNR = 1000. However, the numbers don't agree. What is the reason for the difference?

Section 4: What are the expected uncertainties of the validation data products used – the text talks about reasonable agreement, but there are differences of several km's in some cases, so it would be useful to know if that can be explained by uncertainties in the other datasets as well? In particular, for Kasatochi, the quoted values for prior OMI retrievals are a few km's lower – has the reason for this been looked at in more detail?

Figures 4, 5, 7: These would be clearer if all the instruments were plotted on the same colour scale and lat/lon range. Also, if possible, replot the Caliop data to focus on the relevant region. Similarly for Figure 8, it would be clearer if all the instruments were plotted on the same axes.

In Figures 7 and 8 for the Raikoke eruption, the distribution of values for OMI and TROPOMI seem to be mirrored e.g. OMI has a tail of lower values while TROPOMI has a tail of higher values. Is there an explanation for this?

Section 4.3 – the first paragraph reads as an introduction to section 4.2 too – should the ordering be changed?

Section 4.5 'Discussion of errors'. Have the authors looked in any more detail on the impact of some of their assumptions in the radiative transfer modelling on the retrieval errors? E.g. they mention that using a fixed solar irradiance spectrum will be less

accurate than using the OMI solar measurements. Has the expected impact on this been quantified? Is a fuller assessment of these sorts of errors planned as part of their future work?

Conclusion: Line 472 "with absolute errors of up to 1.5km" – This seems to be the first time that number is quoted, and given the uncertainties and difficulties comparing instruments it may be too strong to put a hard number on an absolute error e.g. The section on errors just mentions 1-2km differences. I think if it's quoted like this, it would be good to back it up with more quantitative information as to where it came from. Otherwise, I would rephrase. (Similarly, the abstract quotes errors of 1-1.5 km's, which should also be made consistent).

Why are the figures 1A and 2A supplemental, as they're directly referenced in the text?

Technical corrections:

- Some abbreviations need defining (line 61: VCD, line 93: BUV)

- Some of the figure numbering is wrong (Line 157, 159 – should reference fig 2a,2b,2c)

- Line 164 – 'absorption' repeated.

––––––––––––––––––––

---

## Author Comment (AC1) · 28 Jan 2021

The comment was uploaded in the form of a supplement:
https://amt.copernicus.org/preprints/amt-2020-376/amt-2020-376-AC1-supplement.pdf

———————————————————

---

## Author Comment (AC2) · 28 Jan 2021

**Response to Comments by Anonymous Referee #2**

We thank the reviewer for taking the time to review the manuscript and provide comments. We agree that there are areas that need clarifications. Suggestion for changes in figures was also much appreciated. The reviewer comments are highlighted in bold with our responses written below. We also state the changes that will be made in the revised manuscript with regard to each comment.

**1) The last sentence of the abstract states "This approach offers a promising prospect of using physics-based machine learning applications to other instruments." However, as this algorithm did not originate in this study or for the OMI instrument, and so has already been demonstrated for a number of other instruments, as phrased here the statement doesn't seem valid. I would suggest the sentence is either rephrased (if that wasn't it's intended meaning), or removed.**

We agree that the statement is not necessary and have removed it from the abstract text.

**2) The section describes IASI retrievals of SO2 height, and follows it with the phrase ('For these techniques, extensive radiative transfer modeling is needed: : :). However, in the Clarisse 2014 paper referenced, this describes a fast retrieval scheme, where this statement may not follow. I would suggest checking and amending the text appropriately.**

It is true that the 2014 paper discusses an updated IASI retrieval that is much faster, therefore the 2014 reference should not be included after that sentence. We have included another sentence that highlights the retrievals from the 2014 paper separately.

**3) Section 2.4 Line 264-266: "The output is a predicted SO2 layer height based on the input of a radiance spectra and associated parameters, including VZA, SZA, RAA, surface albedo and surface pressure, for a single OMI pixel." Is this sentence in the right place – I found the flow of the paragraph a bit confusing, as it then jumps back to talking about convolving the irradiance spectra and then applying PCA?**

Yes, we agree that sentence was out of place and a bit redundant. It has been removed from text.

**4) Section 2.4, Line 278: The text assumes that readers will be familiar with the OMI row anomaly, which may not be the case – it would be useful to explain this somewhere.**

We acknowledge that we did not provide background information on the row anomaly. We added an introduction to the row anomaly in Section 1 when introducing the OMI instrument. The cause of the anomaly is also indicated.

**5) Section 3: Tables 2 and 3 need more explanation in the text here e.g. RMSE is mentioned, but it's not explicitly stated what this represents anywhere. Also, from**

**their captions, I would have expected the RMSE numbers in Table 2 for VCD > 40 and SZA**
**< 75, to be the same as the RMSE in Table 3 for SNR = 1000. However, the numbers don't agree. What is the reason for the difference?**

- Thank you for pointing this out. It seems that the noise analysis was performed earlier with different neural network conditions, as those were changed multiple times when trying to optimize the training. We have redone analysis for Table 3 (below) using the same NN setup and test dataset as for the Table 2 which has resolved the discrepancy.

**Table 3:** The RMSE and the mean absolute difference of all data points in the independent test set after adding noise as indicated by different SNR values. All other parameters and input data were kept constant. SZA < 75 degrees and SO2 VCD > 40 DU were excluded from the test set for these comparisons.

|  | No noise | SNR=1000 | 750 | 500 | 200 | 100 |
|---|---|---|---|---|---|---|
| Mean Absolute Difference (y_known - y_pred) (km) | 0.894 | 0.904 | 0.939 | 0.996 | 1.114 | 1.362 |
| RMSE (km) | 1.454 | 1.498 | 1.521 | 1.632 | 1.807 | 2.143 |
| R-coefficient | 0.988 | 0.985 | 0.983 | 0.980 | 0.972 | 0.955 |

- On the second point, we will be sure to explain Table 2 and 3 better in the text. The RMSE was a metric used to evaluate neural network performance, more specifically the error difference between the "predicted" height (i.e. output from NN based on test data) and "actual" height which is the output from the training set that was used in training.

**6) Section 4: What are the expected uncertainties of the validation data products used - the text talks about reasonable agreement, but there are differences of several km's in some cases, so it would be useful to know if that can be explained by uncertainties in the other datasets as well? In particular, for Kasatochi, the quoted values for prior OMI retrievals are a few km's lower – has the reason for this been looked at in more detail?**

For TROPOMI retrieval (Hedelt et al., 2019) there is a stated retrieval uncertainty of < 2km for $SO_2$ column of greater than 20 DU. However, this is only for the retrieval using the synthetic data. Using real data also adds a certain degree of error. The IASI retrieval also contains an uncertainty of 2 km. In some cases there is more than 2 km difference between different datasets. In addition to uncertainties within validation datasets there are also differences between retrieval technique which could also add to the differences.

The previous OMI $SO_2$ height retrieval pertaining to Kasatochi is from Yang et al, 2010. The height values for Kasatochi were found to be around 9-11 km with uncertainty of up to 2 km as well. We believe this is within reasonable range of our retrieval, especially taking the difference

of retrieval technique into account. We agree that this is important to discuss, and we have added a few sentences in the discussion clarifying this.

> **7) Figures 4, 5, 7: These would be clearer if all the instruments were plotted on the same colour scale and lat/lon range. Also, if possible, replot the Caliop data to focus on the relevant region. Similarly for Figure 8, it would be clearer if all the instruments were plotted on the same axes.**

Thank you for pointing this out. We have replotted figures so color scales and coordinate ranges match. Figure 4c has also been updated using GOME-2 data.

For the CALIPSO lidar data, we agree that it would be helpful to focus in on the region with the volcanic plume. The figures have been replotted for the revised manuscript. As an example here is one for Raikoke (June 23rd, 2019) which would correspond to Figure 9a.

[Figure]

> **8) In Figures 7 and 8 for the Raikoke eruption, the distribution of values for OMI and TROPOMI seem to be mirrored e.g. OMI has a tail of lower values while TROPOMI has a tail of higher values. Is there an explanation for this?**

That is a good observation. It would be difficult to pin point the exact reason because of instrument and retrieval difference, but one interesting observation was that radiance measurement from the instruments were obtained from different cross track positions (rows).In other words the retrievals for OMI were on the left side of the swath (high to low VZA) while for TROPOMI it was the opposite side of the its swath (low to high). Since VZA was one of the parameters involved in training process, this could have some effects on the mirrored distribution. The maps below show VZA plotted for the $SO_2$ plume area.

[Figure]

However, we do not think that there should be a big dependence in the retrieval on VZA. Another explanation could be differences in signal to noise ration (SNR) at nadir (VZA ~ 0) versus the edges where there can be some degradation. There are also differences in SNR between OMI and TROPOMI and furthermore, TROPOMI spectra in the UV is affected by instrument degradation issues. We have included a few sentences about this in Section 4 that discuss these points.

**9) Section 4.3 – the first paragraph reads as an introduction to section 4.2 too – should the ordering be changed?**

That is a good catch. This paragraph was moved to the beginning of section 4.2 instead.

**10) Section 4.5 'Discussion of errors'. Have the authors looked in any more detail on the impact of some of their assumptions in the radiative transfer modelling on the retrieval errors? E.g. they mention that using a fixed solar irradiance spectrum will be less accurate than using the OMI solar measurements. Has the expected impact on this been quantified? Is a fuller assessment of these sorts of errors planned as part of their future work?**

Yes, quantifying the effect of instrument versus the fixed spectrum is more for future work. In general it makes sense that using the irradiance measurements from the same instrument would carry less potential error, however there are benefits for using a fixed reference spectrum. The downside in using instrument irradiance is that the solar irradiance spectrum varies month to month and for each row. This would require far more computation and additional complexity in training thus making it less suitable for future NRT implementations. Additionally we consider applying this algorithm to other instruments, which with a fixed irradiance spectrum is possible without having to repeat the calculations of synthetic radiances. We will remove the accuracy comparison from the manuscript since it is not backed up quantitatively at the moment

There are also many aspects of both the radiative transfer modeling and neural network that can be explored more in depth (i.e. sensitivity analyses). Our primary goal was to obtain a robust algorithm and reasonably accurate algorithm first, however we certainly plan to explore certain sensitivities as future work and if they make a significant impact on result.

**11) Why are the figures 1A and 2A supplemental, as they're directly referenced in the text?**

We initially placed them in supplemental section since they are finer details of the methodology. However, since they ended up being referenced, we will move them to main figures and adjust figure numberings accordingly.

**12) Conclusion: Line 472 "with absolute errors of up to 1.5km" – This seems to be the first time that number is quoted, and given the uncertainties and difficulties comparing instruments it may be too strong to put a hard number on an absolute error e.g. The section on errors just mentions 1-2km differences. I think if it's quoted like this, it would be good to back it up with more quantitative information as to where it came from. Otherwise, I would rephrase. (Similarly, the abstract quotes errors of 1-1.5 km's, which should also be made consistent).**

We changed the statement to "1-2 km" since this is the likely range of errors. You are correct in pointing out that stating an exact error value is not valid here.

---

## Author Response (AR1)

The reviewer comments are highlighted in bold with our responses written below. We state the specific changes made in the revised manuscript in red

**Response to Comments by Anonymous Referee #1**

We thank the reviewer for taking the time to provide feedback and comments on this manuscript. There were many insightful comments concerning the machine learning aspect of this study. We agree that we can conduct more in-depth investigation on the sensitivity analyses and NN parameters. As these details could be of interest to many readers, we aim to take these comments into account and add descriptions of the machine learning into the manuscript along with some into the supplemental material. There are also some good suggestions of potential improvements in the future that we feel may take a lot more additional time to perform and would be more appropriate for future follow-up studies.

1) **Line 194-195: How large is the sample size large enough? I think it would be good to have the number of trainable parameters in the DNNs reported here.**

To check sensitivity to sample size we performed a simple test in reducing the number of training samples by different amounts (e.g. 10%, 20%, 50%) and seeing how this affects test set prediction. While there was a noticeable effect after 50% reduction where the absolute error went up by around 0.3 km, when reducing by 10% the error stayed the same at around 1 km. The RMSE also goes up after dropping more than 10% of data. Given there's very little change when reducing number of samples by 10%, we could expect to see diminishing returns in retrieval quality with more samples.

| % withheld | 0 | 10 | 20 | 30 | 40 | 50 |
|---|---|---|---|---|---|---|
| Mean Abs Difference | 0.95 | 0.98 | 1.02 | 1.08 | 1.12 | 1.24 |
| RMSE | 1.46 | 1.45 | 1.62 | 1.69 | 1.79 | 2.00 |

We have added this discussion to the revised paper at Line 272-280.
Table added to appendix as Table A1.

2) **Line 244-246: Have you tried to increase the number of neurons in each layer of the DNNs or increase the number of hidden layers in the architecture? A sensitivity analysis during a "fine-tuning" process is very helpful for the optimization of DNNs.**

Yes, we have tried to changing the  number of neurons and the number of hidden layers and looking at sensitivities in the results. In our case, less complex NN structure led to more stability in the NN in the application to real data. Using larger amounts of neurons sometimes added unrealistic spikes in the results. The TROPOMI retrieval (Hedelt et. al, 2019) also had a simple two layer NN structure, with 32 and 10 neurons. In the future, we plan to  attempt to further optimize and fine tune the NNs in the retrieval setup.

3) **Line 233-234: The tanh activation function is more frequently used in classification problems. For regressions, the rectified linear unit (ReLU) and parametric ReLU (PReLU) are more used.**

Thank you for pointing this out. We tested ReLU as an activation function, but the tanh seemed to perform better. We demonstrated this with an additional sensitivity analysis (below) which shows that tanh produces less error between the trained NN and test data set. The errors were determined by taking average of 5 training runs since training can slightly vary each time and the process was repeated for multiple datasets (OMI row based). In general, even though the ReLU does not reduce the performance by a lot, in our case using it does not improve the results. PReLU also did not improve the mean error. Note that for this analysis the random seeds and test data were held constant.

Mean absolute error (in km) between test set and predicted outputs by trained NN for different activation functions:

|  | Row 2 | Row 9 | Row 18 | Row 27 |
| --- | --- | --- | --- | --- |
| tanh | 0.968 | 0.943 | 1.035 | 1.036 |
| ReLU | 1.168 | 1.1135 | 1.154 | 1.174 |
| PReLU | 1.097 | 1.064 | 1.094 | 1.107 |

4) **Line 217: How do you control overfitting? Do you have a validation set? A validation set should be a small subset sampled randomly from the training set, over which the performance of the DNNs is checked after each epoch of the training. Sometimes the training loss gets reduced, but validation loss is not.**

- For overfitting, we chose to use L2 regularization. This has now been made clear in the revised manuscript

The issue of overfitting and the L2 Regularization technique are addressed on Line 260-265.

- During the NN training the dataset is further split into a test and validation set with a 0.9/0.1 split. This validation set is different from the independent "test" set that is withheld from training. We added this information to the manuscript.

Sentences added at Lines 246-249.

- Lastly, we did make sure that the training loss of the validation set decreased at a similar rate as the training set and did not increase, in order to avoid overfitting. The training was stopped when validation loss was relatively constant for at least 30 epochs.

5) **Figure 3: Relative errors are much more meaningful here because the "truth" of volcanic SO2 layer heights is not a constant set. It is also clear that the predictability of volcanic SO2 layer heights has a strong correlation with SO2 column information. Since OMI is biased, have you tried to add SO2/O3 column information from other sources as predictors?**

This is an interesting idea and we have not attempted it thus far. The difficulty in using column amounts from other instruments is that other instruments (TROPOMI, OMPS, etc.) have different overpass times, spatial resolutions and number of cross track positions. The application phase retrievals are done on a row/pixel basis, meaning each input sample includes the radiances + parameters for that given pixel. The difference in resolutions and overpass times would make it tricky to be included with OMI data in terms of data processing. Secondly, there is a strong correlation between $SO_2$ height and $SO_2$ amount regardless of instrument as column amount algorithms typically require an assumed profile in the first place. It may be possible to add $O_3$ columns from an assimilated model dataset, however $SO_2$ columns from the models strongly depend on the input of emissions from volcanic eruptions, which in turn are often largely constrained by satellite observations.

6) **Line 229-231: If you scaled the parameters to be within the max/min range, plus the tanh activation function you used, the gradients of DNNs with respect to trainable parameters in DNNs would not be sensitive to predictors closer to the max/min values. The degraded performance of your DNNs shown in Figure 3b could be a result of this. Maybe try batch normalization?**

- Figure 3b shows degradation at high SZAs. Aside from NN parameters, this is also explained by worsening performance of the radiative transfer calculations at high SZAs (strongly reduced signals due to light absorption leading to much lower signal-to-noise ratios).
- Thanks for pointing out the possible effect of the max/min scaling. We agree this can pose a problem, although in our case satellite observations tend to have a much smaller range in the parameters for a given area. Therefore this is not expected to be a big problem when applying to OMI for the volcanic cases in the paper. We still hope to fix this issue while further optimizing the algorithm in the future.

7) **Regarding the stability of DNNs, you could also consider to add skip connections. This is not technically hard. It would smooth the surface of the loss function and reduce the number of local minima. (arXiv:1712.09913)**

Thank you for the suggestion. As previously mentioned, we chose to use L2 regularization as the main method of improving stability and avoiding overfitting. This seemed to work adequately well. The "Dropout" technique was considered, however, it performed worse than L2 regularization for our problem. We will consider the skip connections in future work when we may attempt to optimize the algorithm, but it does not seem like a simple implementation in the Python Keras module that we used for this particular study.

8) **It is difficult to evaluate the performance of your DNNs by comparisons shown in Figures 4, 5, 7 and 9. A heat map could be very helpful here, between the predicted volcanic SO2 layer heights and those retrieved from other satellites.**

We appreciate the suggestion. Comparing the satellite retrievals spatially present an issue since the overpass of OMI is not the same as the other instruments. Therefore the movement of the plume in between measurements needs to be taken into account. To get a general idea of the agreement we included the PDF plots as the comparison. It is also worth noting that the techniques for retrieving have some differences which can influence distribution of values within the plumes and that the information content and sensitivity of IASI IR retrievals differ from UV based retrievals, contain different physical parameters.

9) **Table 2 and 3 show statistics intercompared within the synthetic data set. I think it is meaningful to also have statistics compared with other independent satellite retrievals.**

For Table 2 and 3 the main goal was to show mainly the sensitivities within the NN training based on noise and restricting certain parameters. Comparing statistics between other retrievals in the application stage is also possible but with different metrics such as mean, median, quartiles etc. This is somewhat illustrated by the PDFs.

We have implemented the suggestion to include another table with quantitative comparisons of the retrievals in the manuscript (Table 4). Metrics include standard deviation, mean, median and inner quartile range. Text referencing the table has been added in revised manuscript at Lines 361-363 and Lines 424-428.

10) **If you do have access to other retrievals of volcanic SO2 layer heights, then I would suggest a multi-stage training. In stage 1, synthetic data sets are used. In state 2, you can keep training the model from stage 1, using a subset of real retrievals as the training set and the other as the testing set. If only synthetic data sets are used and the forward model generating these sets is not perfect, then the trained model must have a degraded performance compared to the forward model (due to errors from deep learning model itself). Moreover, during the forward model calculation, if the column data (O3 and SO2) are sampled within some range, then your DNNs would have a difficult time in predicting outliers and extrapolation should be very careful. Speed would be the only advantage then. The data-driven nature of DNNs should be taken advantage of.**

- Similar to using column amounts from other satellite data sets, introducing other height retrieval would also add error due to differences in overpass time.
- Another main challenge is the lack of fast and reliable retrievals of $SO_2$ layer height. Furthermore IASI (infrared) retrievals should not ideally be used together with OMI

outside of a result comparison since some physical parameters are different between the two.

- It is true that there would be issues with extrapolating far outside the range of $O_3$ and $SO_2$ column. However, in volcanic eruptions the $SO_2$ column rarely exceeds 1000 DU and ozone column is also within the range used for forward RT calculations. If necessary, additional spectra can be calculated with an extended range of parameters and included in training.

11) **Bayesian neural networks are ideal for such prediction problem, when there are uncertainties associated with the outputs. This could be future work due to the technical complexity. But you could still alter the random seed to generate an ensemble of DNNs, such that you can provide a rough estimate of uncertainties on the predictions.**

- This is a very interesting idea. Ensembles of DNNs may improve the performance (accuracy) of single networks for solving remote sensing problems (Loyola, 2006), usage of ensembles for the $SO_2$ layer height retrieval will be investigated in the future.

Sentence on this added in the Conclusion section at Lines 528-530.

- Thank you for this suggestion on altering random seed. We have attempted to change random seeds in the NN for initializing the weights and biases, but this did not impact $SO_2$ height results significantly with our current NN setup. We agree it may be useful to use the random seed variation to produce an estimate of random error within the NN. This idea has been implemented and tested for the training phase and on two of the OMI orbits to obtain an uncertainty measure of changing random seeds within the machine learning.

This has been included as a table of error values (Table A2) in the revised paper. Text discussion is in Section 4.5 at Lines 474-481.

**Minor comments**
**Line 61: replace VCD by vertical column density (VCD)**

Changed in manuscript (line 61-62).

**Fig 4: can you change the colorbar of (c) to the same scale as (a) and (b)?**

We have replotted Figure 4c to match the same color scales/plot format.

**Response to Comments by Anonymous Referee #2**

We thank the reviewer for taking the time to review the manuscript and provide comments. We agree that there are areas that need clarifications. Suggestion for changes in figures was also much appreciated. We also state the changes that will be made in the revised manuscript with regard to each comment.

**1) The last sentence of the abstract states "This approach offers a promising prospect of using physics-based machine learning applications to other instruments." However, as this algorithm did not originate in this study or for the OMI instrument, and so has already been demonstrated for a number of other instruments, as phrased here the statement doesn't seem valid. I would suggest the sentence is either rephrased (if that wasn't it's intended meaning), or removed.**

We agree that the statement is not necessary and have removed it from the text (Line 34-35).

**2) The section describes IASI retrievals of SO2 height, and follows it with the phrase ('For these techniques, extensive radiative transfer modeling is needed: : :). However, in the Clarisse 2014 paper referenced, this describes a fast retrieval scheme, where this statement may not follow. I would suggest checking and amending the text appropriately.**

It is true that the 2014 paper discusses an updated IASI retrieval that is much faster, therefore the 2014 reference should not be included after that sentence.

We have included another sentence that highlights the retrievals from the 2014 paper separately (Lines 112-114).

**3) Section 2.4 Line 264-266: "The output is a predicted SO2 layer height based on the input of a radiance spectra and associated parameters, including VZA, SZA, RAA, surface albedo and surface pressure, for a single OMI pixel." Is this sentence in the right place – I found the flow of the paragraph a bit confusing, as it then jumps back to talking about convolving the irradiance spectra and then applying PCA?**

Yes, we agree that sentence was out of place and redundant.
It has been removed from text at Line 289-291 of marked version.

**4) Section 2.4, Line 278: The text assumes that readers will be familiar with the OMI row anomaly, which may not be the case – it would be useful to explain this somewhere.**

We acknowledge that we did not provide background information on the row anomaly.

We added an introduction to the row anomaly on lines 71-75 in Section 1 when introducing the OMI instrument. The cause of the anomaly is also indicated.

**5) Section 3: Tables 2 and 3 need more explanation in the text here e.g. RMSE is mentioned, but it's not explicitly stated what this represents anywhere. Also, from**

**their captions, I would have expected the RMSE numbers in Table 2 for VCD > 40 and SZA**
< 75, to be the same as the RMSE in Table 3 for SNR = 1000. However, the numbers don't agree. What is the reason for the difference?

- Thank you for pointing this out. It seems that the noise analysis was performed earlier with different neural network conditions, as those were changed multiple times when trying to optimize the training. We have redone analysis for Table 3 (below) using the same NN setup and test dataset as for the Table 2 which has resolved the discrepancy.

**Table 3:** The RMSE and the mean absolute difference of all data points in the independent test set after adding noise as indicated by different SNR values. All other parameters and input data were kept constant. SZA < 75 degrees and SO2 VCD > 40 DU were excluded from the test set for these comparisons.

|  | No noise | SNR=1000 | 750 | 500 | 200 | 100 |
|---|---|---|---|---|---|---|
| Mean Absolute Difference (y_known - y_pred) (km) | 0.894 | 0.904 | 0.939 | 0.996 | 1.114 | 1.362 |
| RMSE (km) | 1.454 | 1.498 | 1.521 | 1.632 | 1.807 | 2.143 |
| R-coefficient | 0.988 | 0.985 | 0.983 | 0.980 | 0.972 | 0.955 |

- The RMSE was a metric used to evaluate neural network performance, more specifically the error difference between the "predicted" height (i.e. output from NN based on test data) and "actual" height which is the output from the training set that was used in training.

We have explained the errors in Table 2 and 3 in the text at Lines 210-212 and Lines 316-318 respectively.

**6) Section 4: What are the expected uncertainties of the validation data products used - the text talks about reasonable agreement, but there are differences of several km's in some cases, so it would be useful to know if that can be explained by uncertainties in the other datasets as well? In particular, for Kasatochi, the quoted values for prior OMI retrievals are a few km's lower – has the reason for this been looked at in more detail?**

For TROPOMI retrieval (Hedelt et al., 2019) there is a stated retrieval uncertainty of < 2km for $SO_2$ column of greater than 20 DU. However, this is only for the retrieval using the synthetic data. Using real data also adds a certain degree of error. The IASI retrieval also contains an uncertainty of 2 km. In some cases there is more than 2 km difference between different datasets. In addition to uncertainties within validation datasets there are also differences between retrieval technique which could also add to the differences.

Added discussion in Section 4 on Lines 474-481.

The previous OMI SO$_2$ height retrieval pertaining to Kasatochi is from Yang et al, 2010. The height values for Kasatochi were found to be around 9-11 km with uncertainty of up to 2 km as well. We believe this is within reasonable range of our retrieval, especially taking the difference of retrieval technique into account.

We agree that this is important to address, and we have clarified this in Section 4.1, Lines 353-355.

**7) Figures 4, 5, 7: These would be clearer if all the instruments were plotted on the same colour scale and lat/lon range. Also, if possible, replot the Caliop data to focus on the relevant region. Similarly for Figure 8, it would be clearer if all the instruments were plotted on the same axes.**

Thank you for pointing this out.
Figure 4c has been replotted with the GOME-2 data. We have also replotted Figures 5 and 7 so all panels have matching color scales and coordinate ranges.

For the CALIPSO lidar data, we agree that it would be helpful to focus in on the region with the volcanic plume.

All CALIPSO data (Figure 6d, 7e, 7f, 11a and 11b) have been replotted for the revised manuscript.

**8) In Figures 7 and 8 for the Raikoke eruption, the distribution of values for OMI and TROPOMI seem to be mirrored e.g. OMI has a tail of lower values while TROPOMI has a tail of higher values. Is there an explanation for this?**

That is a good observation. It would be difficult to pin point the exact reason because of instrument and retrieval difference, but one interesting observation was that radiance measurement from the instruments were obtained from different cross track positions (rows).In other words the retrievals for OMI were on the left side of the swath (high to low VZA) while for TROPOMI it was the opposite side of the its swath (low to high). Since VZA was one of the parameters involved in training process, this could have some effects on the mirrored distribution. The maps below show VZA plotted for the SO$_2$ plume area.

[Figure]

However, we do not think that there should be a big dependence in the retrieval on VZA. Another explanation could be differences in signal to noise ratio (SNR) at nadir (VZA ~ 0) versus the edges where there can be some degradation. There are also differences in SNR between OMI and TROPOMI and furthermore, TROPOMI spectra in the UV is affected by instrument degradation issues.

We have pointed out this issue in the revised manuscript in Section 4, Lines 457-461.

**9) Section 4.3 – the first paragraph reads as an introduction to section 4.2 too – should the ordering be changed?**

Thank you for pointing this out.

We slightly modified beginning text of section 4.2 and 4.3 so that each case is introduced separately.

**10) Section 4.5 'Discussion of errors'. Have the authors looked in any more detail on the impact of some of their assumptions in the radiative transfer modelling on the retrieval errors? E.g. they mention that using a fixed solar irradiance spectrum will be less accurate than using the OMI solar measurements. Has the expected impact on this been quantified? Is a fuller assessment of these sorts of errors planned as part of their future work?**

Yes, quantifying the effect of instrument versus the fixed spectrum is more for future work. In general it makes sense that using the irradiance measurements from the same instrument would carry less potential error, however there are benefits for using a fixed reference spectrum. The downside in using instrument irradiance is that the solar irradiance spectrum varies month to month and for each row. This would require far more computation and additional complexity in training thus making it less suitable for future NRT implementations. Additionally we consider applying this algorithm to other instruments, which with a fixed irradiance spectrum is possible without having to repeat the calculations of synthetic radiances.

We removed the "accuracy" statement since there is no quantitative measure. We included a few sentences about the other points at Lines 177-182.

There are also many aspects of both the radiative transfer modeling and neural network that can be explored more in depth (i.e. sensitivity analyses). Our primary goal was to obtain a robust algorithm and reasonably accurate algorithm first, however we certainly plan to explore certain sensitivities as future work and if they make a significant impact on result.

11) **Why are the figures 1A and 2A supplemental, as they're directly referenced in the text?**

We initially placed those figures in supplemental section since they are finer details of the methodology.

Since they ended up being referenced, we have moved them to main figures and adjusted figure numberings accordingly.

12) **Conclusion: Line 472 "with absolute errors of up to 1.5km" – This seems to be the first time that number is quoted, and given the uncertainties and difficulties comparing instruments it may be too strong to put a hard number on an absolute error e.g. The section on errors just mentions 1-2km differences. I think if it's quoted like this, it would be good to back it up with more quantitative information as to where it came from. Otherwise, I would rephrase. (Similarly, the abstract quotes errors of 1-1.5 km's, which should also be made consistent).**

You are correct in pointing out that stating an exact error value is not valid here.

We changed the statement to "1-2 km" (Line 514), since this is the likely range of errors.